# Physiologic biomechanics enhance reproducible contractile development in a stem cell derived cardiac muscle platform

Yao-Chang Tsan[1,2], Samuel J. DePalma [3], Yan-Ting Zhao[4], Adela Capilnasiu[3], Yu-Wei Wu [5], Brynn Elder [1], Isabella Panse[1], Kathryn Ufford [1], Daniel L. Matera [3], Sabrina Friedline[1], Thomas S. O'Leary[6], Nadab Wubshet[7], Kenneth K. Y. Ho [7], Michael J. Previs[6], David Nordsletten[3,8], Lori L. Isom [4,9,10], Brendon M. Baker [3], Allen P. Liu [3,7,11] & Adam S. Helms [1✉]

Human pluripotent stem cell-derived cardiomyocytes (hPSC-CMs) allow investigations in a human cardiac model system, but disorganized mechanics and immaturity of hPSC-CMs on standard two-dimensional surfaces have been hurdles. Here, we developed a platform of micron-scale cardiac muscle bundles to control biomechanics in arrays of thousands of purified, independently contracting cardiac muscle strips on two-dimensional elastomer substrates with far greater throughput than single cell methods. By defining geometry and workload in this reductionist platform, we show that myofibrillar alignment and auxotonic contractions at physiologic workload drive maturation of contractile function, calcium handling, and electrophysiology. Using transcriptomics, reporter hPSC-CMs, and quantitative immunofluorescence, these cardiac muscle bundles can be used to parse orthogonal cues in early development, including contractile force, calcium load, and metabolic signals. Additionally, the resultant organized biomechanics facilitates automated extraction of contractile kinetics from brightfield microscopy imaging, increasing the accessibility, reproducibility, and throughput of pharmacologic testing and cardiomyopathy disease modeling.

[1] Division of Cardiovascular Medicine, University of Michigan, Ann Arbor, MI, USA. [2] Department of Human Genetics, University of Michigan, Ann Arbor, MI, USA. [3] Department of Biomedical Engineering, University of Michigan, Ann Arbor, MI, USA. [4] Department of Pharmacology, University of Michigan, Ann Arbor, MI, USA. [5] Institute of Molecular Biology, Academia Sinica, NanKang, Taipei, Taiwan. [6] Molecular Physiology and Biophysics, University of Vermont, Burlington, VT, USA. [7] Department of Mechanical Engineering, University of Michigan, Ann Arbor, MI, USA. [8] Department of Cardiovascular Surgery, University of Michigan, Ann Arbor, MI, USA. [9] Department of Neurology, University of Michigan, Ann Arbor, MI, USA. [10] Department of Molecular and Integrative Physiology, University of Michigan, Ann Arbor, MI, USA. [11] Department of Biophysics, University of Michigan, Ann Arbor, MI, USA. ✉email: adamhelm@umich.edu

Modern techniques enable the efficient generation of cardiac muscle cells from human pluripotent stem cells (hPSC-CMs), and studies with these cells have provided important insights into human cardiac development and disease[1–3]. However, reproducible assessment of contractile function, the most fundamental property of heart muscle, has remained a major challenge due to the immaturity of hPSC-CMs on standard 2D substrates, where they innately exhibit reduced membrane voltages, disorganized myofibrils, and weak contractile forces[4,5]. For example, motion analysis and/or sarcomere shortening in traditionally cultured hPSC-CMs suffers from extensive heterogeneity due to the lack of myofibrillar organization[6–8]. Single-cell micropatterning can control cell shape in 2D and enables characterization of single-cell traction forces[9,10], but is limited by low throughput, marked variability among replicates, and a lack of intercellular junctions critical to normal cardiac physiology[11]. Three-dimensional (3D) cardiac tissues suspended between elastomer posts allow physiologic uniaxial contractions and induce the highest level of iPSC-CM maturation to date[5]. While clearly increasing maturation, other challenges to standardized analyses with 3D tissues include (1) requirement for expertise in 3D tissue fabrication, (2) a requirement for stromal cell admixture[12,13] that may amplify heterogeneity and batch-to-batch variation, (3) an inability to perform concurrent imaging of myofibrils due to tissue thickness, and (4) limited throughput[5]. The primary biomechanical drivers of improved development in 3D cardiac tissues are likely the alignment and fractional shortening that occur during coordinated contractions against elastomer substrates, but dissection of individual contributions in this system is challenging due to stromal cell admixture[5]. Thus, a 2D method that could capture the improved biomechanical aspects of 3D tissues would enable a simplified platform for investigating contractile function in iPSC-CMs.

In this work, we developed a reductionist approach to simultaneously test the importance of physiologic biomechanics on cardiac development while also improving the reproducibility of contractile analyses in 2D. In our micron-scale 2D cardiac muscle bundle (2DMB) approach, we use micropatterning of elastic polydimethylsiloxane (PDMS) to anchor thin, purified cardiac muscle strips in 2D arrays with improved attachment yield than has been possible with single cells. In these arrays, individual cardiac muscle bundles have a controlled geometry and contract independently against a defined workload. Using this highly organized biomechanical environment, we show that myofibrillar alignment and physiologic (auxotonic) contractions in an optimized media environment drive marked development of contractile function, calcium handling, and electrophysiology. These physiologic improvements coincide with a developmental shift toward terminally differentiated, nonproliferative cardiomyocytes with an upregulated myofilament growth program that we show is causally linked to contractile force. In addition, using an automated contractile analysis method ("ContractQuant"), we show that the 2DMB method reduces variability across replicates, diminishes the confounding effect of myofibrillar disorganization, and increases statistical power for pharmacologic testing. Finally, we demonstrate that the reproducible contractile microenvironment of 2DMBs allows them to be used as a test bed for investigation of developmental cues that intersect with contractile function, and as a platform for precise dissection of contractile kinetics in cardiomyopathy disease models in a simplified 2D system.

## Results

### Micropatterning of low-modulus PDMS enables cardiac muscle bundle formation.
We recently reported that single micropatterned hPSC-CMs exhibit highly variable and often suboptimal myofibrillar development that contributes to marked replicate variability and low throughput[11]. We hypothesized that this stalled development may be due to a lack of connectivity with adjacent hPSC-CMs. To test this hypothesis, we compared the cell size of isolated hPSC-CMs to hPSC-CMs in contact with other hPSC-CMs. Cardiomyocytes in contact with at least one other cardiomyocyte grew to significantly larger sizes than isolated cells (Supplemental Fig. 1). This finding motivated the development of a platform to generate geometrically defined multicellular cardiomyocyte constructs that harness the benefits of reproducible myofibrillar alignment and uniaxial contractility via micropatterning[9,10,14], while also enabling hPSC-CM connectivity.

Optimal induction of myofibrillar alignment and contractility in single cardiomyocytes is observed with a 7:1 aspect ratio, simulating adult myofiber eccentricity[9,15]. Therefore, we scaled our prior single-cell micropatterning approach[11,14] by 8-fold ($308 \times 45\,\mu m$) for the 2DMB platform (Fig. 1). The design separates individual micropatterns by a buffering distance to mechanically decouple adjacent 2DMBs (4772 patterns covering 26% of the surface area of a 1.6 by 1.6 cm stamp area). We initially tested these micropatterns on elastic polyacrylamide gels[14,16]. However, although hPSC-CMs adhered to micropatterned polyacrylamide, contractile forces invariably resulted in delamination of 2DMBs within 48 h (Supplemental Video 1). In contrast, using a recently described micropatterning technique for low-modulus PDMS[17–19], we found that 2DMBs on 8 kPa PDMS maintained highly consistent adherence. Using differentiation-day 24 hPSC-CMs seeded at a density to achieve 6–12 cells per micropattern, 2DMBs rapidly developed myofibrillar alignment and uniaxial contractions within 3 days post plating and exhibited highly reproducible long-axis orientation of myofibrils by day 8 (Fig. 1a–e and Supplemental Videos 2 and 3). Myofibrils aligned with the long axis in 2DMBs across individual cell junction connection points, as observed with the intercalated disk proteins N-cadherin (Supplemental Fig. 2) and desmoplakin (visualized with live-cell desmoplakin-GFP hPSC-CMs, Supplemental Videos 4 and 5).

To compare myofibrillar alignment in 2DMBs and standard (nonpatterned) monolayer hPSC-CMs, we imaged myofibrils, marked by the sarcomeric protein myosin-binding protein C (MyBP-C), at day 32 and quantified individual sarcomere lengths and orientations. Resting sarcomere length in both 2DMBs and nonpatterned hPSC-CMs on 8 kPa PDMS substrates was ~2.2 μm ($2.2 \pm 0.4\,\mu m$ and $2.2 \pm 0.5\,\mu m$, respectively), both longer than in single hPSC-CMs on PDMS ($1.8 \pm 0.4\,\mu m$, $P < 0.0001$; Fig. 1f). However, nonpatterned hPSC-CMs on 8 kPa PDMS exhibited random orientations of myofibrils, in contrast to the highly reproducible alignment of myofibrils along the long axis of 2DMBs, irrespective of individual cell boundaries (Fig. 1g–j and Supplemental Fig. 2). Taken together, these results show that the 2DMB approach yields highly organized 2D purified cardiomyocyte assemblies with long-axis myofibrillar alignment.

### 2DMBs enable efficient and high-fidelity contractile quantification.
A major barrier to accurate quantification of hPSC-CM contractility on standard 2D substrates has been the heterogeneous myofibrillar orientations and contractile directions (Supplemental Video 6). Since our goal was to mechanically decouple adjacent 2DMBs, we performed a modeling analysis to examine the influence of varying spatial relationships among 2DMBs and the PDMS substrate (Fig. 2a and Supplemental Notes). In 2DMBs across a range of fractional shortening, we found minimal force transmission through the PDMS to edges of the buffering spaces, indicating that 2DMBs are mechanically decoupled from each other (Supplemental Fig. 3). We further

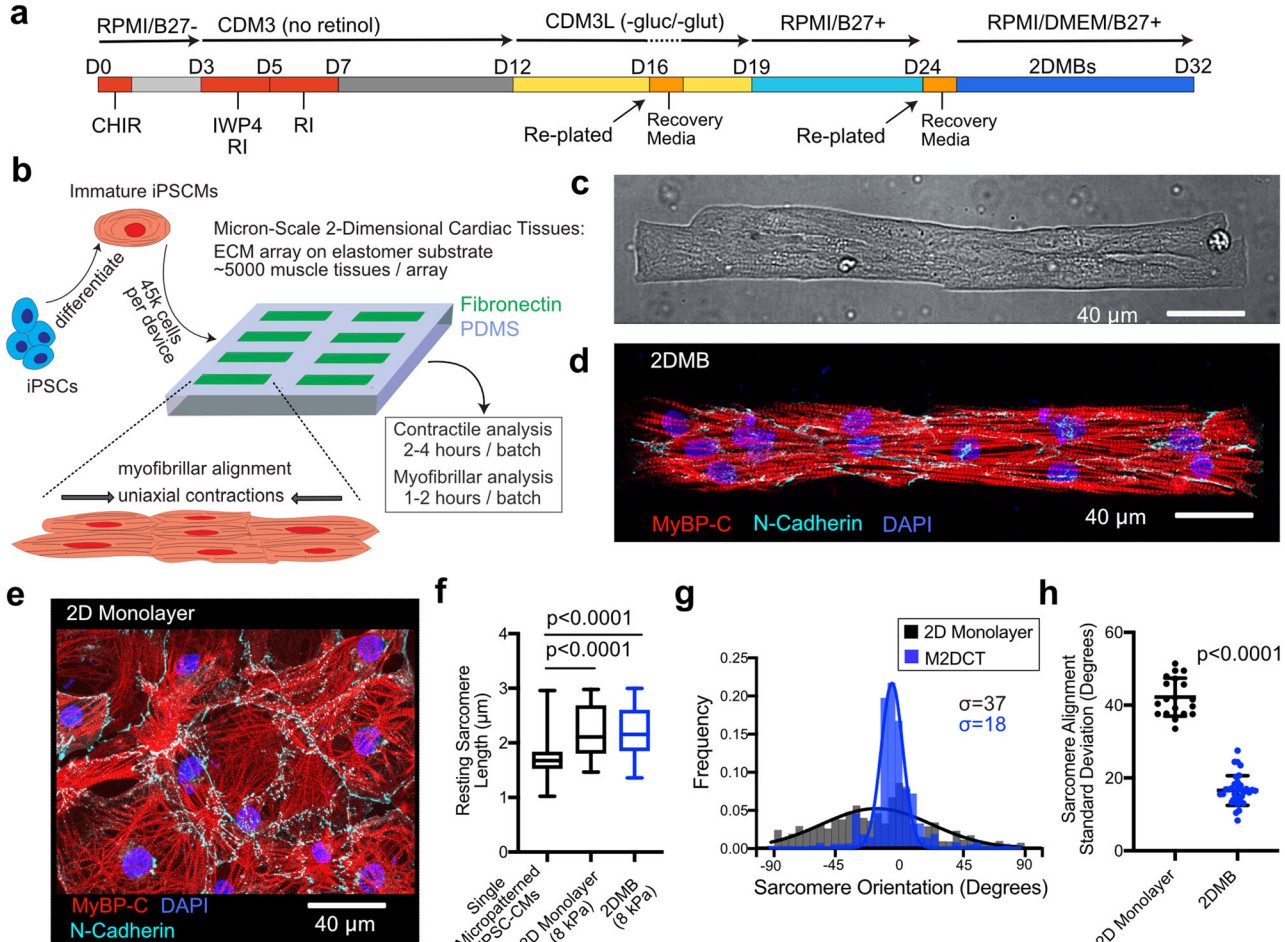

**Fig. 1 Micron-scale two-dimensional cardiac muscle bundles (2DMBs) exhibit enhanced myofibrillar organization. a** iPSCs were differentiated through wnt modulation (CHIR, IWP4) with retinoic signaling inhibition (RI) on days 3–7. During lactate purification, CDM3 was formulated with RPMI without glucose and without glutamine to improve purification. **b** Day-25 hPSC-CMs were replated to 8 kPa PDMS micropatterned with fibronectin in 7:1 aspect ratio, 308 × 45 μm rectangles, each separated by a buffering space to minimize mechanical interactions among adjacent 2DMBs. The overall pipeline improves efficiency by requiring only 45 k hPSC-CMs per substrate, each yielding many high-quality, matured 2DMBs per device for structure and function studies. The approach reduces the overall time required for contractile and myofibrillar structural analyses, including both imaging and running automated image-processing algorithms. **c** hPSC-CMs avidly adhere and conform to PDMS micropatterns but not to surrounding PDMS to generate 2DMBs (image from brightfield microscopy with ×40 objective). **d** Myofibrils (marked by MyBP-C) develop along the 2DMB long-axis traversing continuously across cell junctions (marked by N-Cadherin). hPSC-CM plating density was controlled to attain an average of 6–12 hPSC-CMs per 2DMB. **e** Representative standard (nonpatterned) hPSC-CM multicellular tissue on 8 kPa PDMS exhibits disorganized myofibrils and cell junctions in the lack of directional ECM cues from the substrate. **f** Resting sarcomere length was measured in single micropatterned hPSC-CMs compared to nonpatterned (standard) hPSC-CMs on 8 kPa PDMS and 2DMBs on 8 kPa PDMS (N = 448, 229, 373 sarcomere measurements per condition; box plot: median, interquartile range, range; two-sided Tukey-adjusted ANOVA). **g** Individual sarcomere unit orientations were measured across images using an automated imaging method in MATLAB and were fit to a Gaussian distribution for each image. Representative distribution of sarcomere orientations is shown for an 2DMB (blue) overlayed on a representative distribution for standard hPSC-CMs (black), and the standard deviation (σ) for each is shown. **h** The standard deviation of sarcomere angles is lower for 2DMBs (N = 34) compared to monolayer hPSC-CMs (N = 19) with each distribution compiled from hundreds of individual sarcomere angle measurements; two-sided t test; error bars indicate standard deviation). Source data are provided as a Source Data file.

showed that 8 kPa PDMS completely decouples contracting 2DMBs from the underlying glass at any thickness greater than 40 μm (Supplemental Fig. 3). In addition, we assessed how the cardiomyocyte elastic modulus may influence relationships between fractional shortening and force generation. In the range of cardiac muscle elastic moduli (8–12 kPa)[20], the influence of variability in 2DMBs' intrinsic elastic moduli was minor on 8 kPa PDMS (Supplemental Fig. 3). Finally, we modeled the relationship between fractional shortening measured at different locations within 2DMBs and whole-tissue force to determine optimal locations for the region of interest (ROI) placement when measuring fractional shortening. Based on these results, we established that ROI locations measuring displacements of the inner

50% of the 2DMB length are most robust to minor variations in ROI placement (Supplemental Fig. 3).

Next, we developed a method to quantify contractile function from 2DMBs. To avoid phototoxicity-induced contractile artifacts, we developed the ContractQuant algorithm, which tracks pixel features of ROIs from brightfield time-series images using a cross-correlation method implemented in MATLAB (Fig. 2b–d). The reproducible contractile behavior of 2DMBs enables calculation of contraction and relaxation velocities at multiple phases of each contractile cycle by displacement tracking within ROIs positioned within the inner 50% of 2DMBs (Fig. 2c). MATLAB scripts for ContractQuant are available on Github. Taken together, we show that contractile quantification is highly

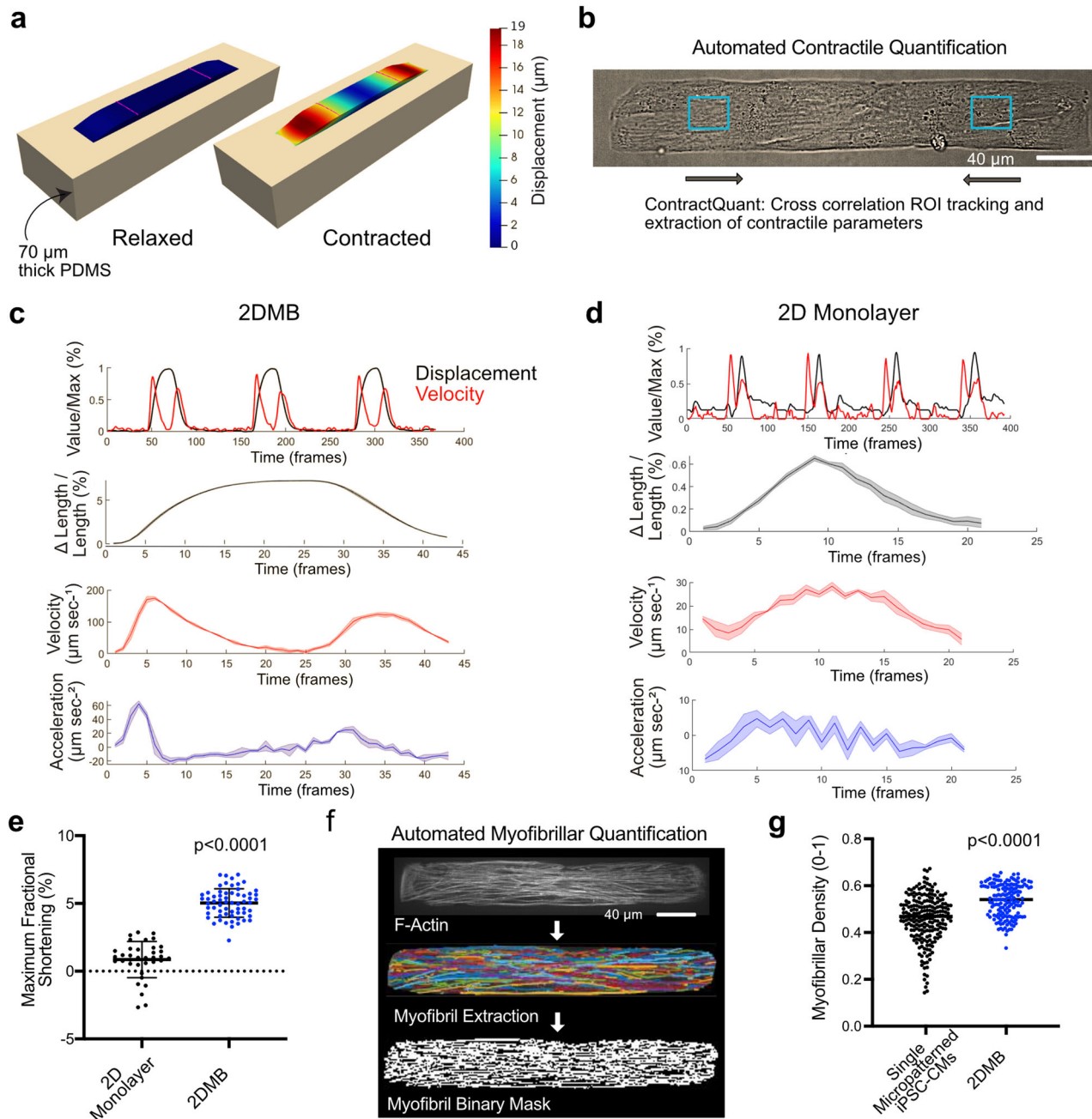

**Fig. 2 Mechanical uncoupling of adjacent micron-scale two-dimensional cardiac muscle bundles (2DMBs) improves contractile function and quantification reproducibility. a** Modeling analysis demonstrated that 2DMBs are mechanically uncoupled from adjacent 2DMBs (see also Supplemental Fig. 3). 3D representation of an 2DMB on 70-μm thick PDMS at 11% fractional shortening shown with heatmap depicting local displacements. Pink lines demarcate the inner 50% length of the 2DMBs where regional muscle bundle deformations are most stable for accurate measurements of fractional shortening. **b** Brightfield image of 2DMB with regions of interest (ROIs) annotated for displacement tracking using cross-correlation in MATLAB with the ContractQuant algorithm. **c** Representative output from ContractQuant for an 2DMB showing (top) concurrent normalized displacements and velocities for the full time-series capture (second row) fractional shortening of merged contractions, (third row) merged absolute values of contractile velocity, and (fourth row) absolute values for acceleration. In rows 2–4, the solid line indicates the mean and shaded regions the 95% confidence intervals. **d** Representative output from ContractQuant for nonpatterned hPSC-CMs on 8 kPa PDMS shows significantly greater heterogeneity across individual contractions that prevents extraction of parameters other than maximal fractional shortening. Note that a different scale was used for Δlength/length (fractional shortening) because of the lower contractile amplitudes in monolayers. In rows 2–4, mean and 95% confidence intervals are shown. **e** Maximal fractional shortening is greater in 2DMBs than standard hPSC-CMs on 8 kPa PDMS ($N = 62$ and 36, respectively). **f** Myofibrils in 2DMBs were imaged (SiR-actin, to label F-actin) following contractile analysis to verify cardiomyocyte purity and homogeneous distribution of myofibrils in analyzed muscle bundles. An automated image analysis pipeline (MyoQuant) was used to extract signal peaks corresponding to individual myofibrillar bundles and estimate the total myofibrillar bundle area. **g** Myofibrillar density (extracted myofibrillar area normalized to cell or 2DMB area using the MyoQuant algorithm) is greater for 2DMBs than for single micropatterned iPSC-CMs. 2DMBs also exhibit less variability ($N = 161$ 2DMBs from seven batches; $N = 219$ single micropatterned iPSC-CMs from 12 batches). *P* values obtained from two-tailed *t* tests; error bars indicate standard deviation. Source data are provided as a Source Data file.

tractable due to the long-axis uniaxial contractile behavior of individual 2DMBs that are mechanically decoupled from each other on a substrate with physiologic elasticity.

**Myofilament function and throughput are improved in 2DMBs**. To compare myofilament function in 2DMBs versus standard monolayer hPSC-CMs, we measured maximal fractional shortening. Reproducible measurements of parameters other than fractional shortening in monolayer hPSC-CMs were not possible due to excessive irregularity in contractile behavior (Fig. 2d). Average maximal fractional shortening was significantly greater in 2DMBs compared to standard hPSC-CMs from the same batches (5.0 ± 1.1% vs. 0.9 ± 1.4%, $P < 0.0001$, Fig. 2e). Moreover, variability across 2DMBs was reduced by 87% compared to standard hPSC-CMs (coefficient of variation 0.21 vs. 1.58, Fig. 2e). In addition, standard hPSC-CMs exhibited regions of apparent stretch due to coupling with adjacent, more strongly contracting cells (Supplemental Video 6).

Micropatterning of single hPSC-CMs overcomes the disorganized myofibrils pervasive in monolayer hPSC-CMs but are low throughput and exhibit stalled myofibrillar development[11]. In our prior work, the yield of single hPSC-CMs meeting quality thresholds for contractile quantification was only 5–36 cells per substrate containing 2742 single-cell-scale rectangular micropatterns (0.5 ± 0.3% of micropatterns, $N = 12$, Supplemental Video 7). In marked contrast, 55 ± 13% of micropatterns on 2DMB substrates exhibited adherence of 6–12 cells with resultant rectangular geometry and uniaxial contractile direction (110-fold increase in yield vs. single cells, $P < 0.0001$, $N = 20$), while a total of 91 ± 5% micropatterns contained at least three hPSC-CMs that exhibited uniaxial contractile direction (Supplemental Video 2). To compare myofibrillar density between single micropatterned hPSC-CMs and 2DMBs, we developed an automated myofibrillar detection algorithm (termed "MyoQuant") that takes as input fluorescence images of F-actin (Fig. 2f)[11]. 2DMBs exhibited denser myofibrils (mean 0.54 ± 0.07 vs. 0.46 ± 0.10, $P < 0.001$) with a reduced coefficient of variation across replicates (0.13 vs. 0.22, $P < 0.001$, Fig. 2g). Mean uniaxial traction force normalized to cell/2DMB width was also greater in 2DMBs vs. single micropatterned hPSC-CMs (122 ± 26 μN mm$^{-1}$ vs 67 ± 30 μN mm$^{-1}$, $P < 0.001$). The person-hours and computational hours required for the 2DMB workflow were markedly reduced compared to single-cell traction force microscopy (Supplemental Fig. 4).

We further demonstrated that the 2DMB technique is generalizable across diverse hPSC lines, including the widely used WTC line and three additional randomly selected hPSC lines from healthy individuals (Supplemental Fig. 5). Since batch-related variability remains a frequent occurrence in hPSC-CM differentiations with current protocols, we hypothesized that the batch-to-batch variations in myofibrillar structure may affect the contractile function in 2DMBs, similar to our prior findings in single micropatterned hPSC-CMs[11]. Consistently, when we measured contractile function concurrently with myofibrillar structure (using MyoQuant[11]) for 150 2DMBs across seven differentiation batches in the DF19-9-11 line, we found that variability in contractile function in 2DMBs was, in part, due to differences in myofibrillar abundance, with a significant correlation between mean myofibrillar area per batch and mean maximal fractional shortening ($R^2 = 0.76$, $P = 0.005$ vs. slope of 0; Supplemental Fig. 6). Combining this live-cell myofibrillar quantification approach with the 2DMB platform enables rapid quality control assessment, including verification of hPSC-CM purity and exclusion of tissues without homogeneous distributions of myofibrils from contractile analyses (Supplemental Fig. 6).

**Electrophysiologic development depends on the biomechanical environment**. Next, we examined whether the improved myofibrillar structure and function associated with the physiologic biomechanical environment of 2DMBs result in maturation of calcium handling, ion channel, and gap junction expression compared to 2D monolayers (Fig. 3a). We found that *SCN5A* and *KCNJ2* were both markedly upregulated in 2DMBs, indicating greater expression of encoded Nav1.5 (responsible for phase 0 depolarization) and Kir2.1 (responsible for maintaining negative membrane potential), respectively. *HCN4*, responsible for sodium influx in pacemaking cells, was downregulated in 2DMBs. *ATP2A2*, *RYR2*, and *CASQ2* genes were significantly upregulated, consistent with increased expression of the sarcoplasmic reticulum proteins, SERCA2, ryanodine-2, and calsequestrin-2, respectively. Finally, the predominant cardiac gap junction gene, *GJA1* (encoding connexin-43), was markedly upregulated in 2DMBs.

To determine whether these gene expression differences correlated with functional improvements, we performed patch clamping in 2DMBs. We reproducibly observed mature resting membrane potentials in 2DMBs (−81 ± 4 mV, Fig. 3b, c), as compared to prior characterizations of immature hPSC-CMs[5]. Analysis of triggered action potentials also revealed a more mature profile—action potential amplitude was 121 ± 4 mV; action potential duration at 90% was 578 ± 88 msec; and maximum dV dt$^{-1}$ (phase 0 depolarization) was 105 ± 31 V sec$^{-1}$ (Fig. 3d–f). We assessed calcium handling with the fura-2 indicator dye. 2DMBs exhibited lower resting calcium levels than standard hPSC-CMs, indicating greater sarcoplasmic reticulum retention consistent with higher *ATP2A2* and *CASQ2* expression (Fig. 3g). Average calcium-transient amplitudes were similar in 2DMBs, but variability across replicates was 73% less in 2DMBs (standard deviation of calcium-transient amplitude 0.29 vs. 1.06 fura-2 ratio, $P < 0.0001$, Fig. 3h). Finally, we quantified gap junction formation, which is essential for mature electromechanical coupling in heart tissue. We imaged connexin-43 using a GFP-reporter line at day 3 and day 10 following 2DMB formation. 2DMBs exhibited greater connexin-43 expression at day 3 compared to standard hPSC-CMs (Fig. 3i), but with predominant perinuclear localization (Fig. 3j), indicating that gap junction reassembly was still underway. Paired analysis of the same 2DMBs imaged at day 10 exhibited a marked increase in connexin-43 abundance with localization at cell–cell borders, as expected for properly localized gap junctions (Fig. 3j).

**2DMBs increase the precision of pharmacologic testing in hPSC-CMs**. Pharmacologic testing is a major application of hPSC-CMs but immaturity and variability of these cells may limit reproducibility in standard 2D monolayers. To test whether the organization of 2DMBs improves reproducibility, we quantified fractional shortening of 2DMBs in the presence of contractile antagonists and agonists. We demonstrated that the contractile inhibitors mavacamten (cardiac-specific myosin ATPase inhibitor) and verapamil (L-type calcium channel inhibitor) both exhibited a highly reproducible concentration-dependent decrement in fractional shortening relative to standard hPSC-CMs (Fig. 4a–d). Increasing media calcium concentration elicited a reproducible concentration-dependent increment in fractional shortening (Fig. 4e). Quantification of contractility in response to mavacamten in 2DMBs from a separate iPSC line also showed a reproducible concentration-dependent response (Fig. 4f). Based on the observed standard deviations, a power analysis revealed that the number of samples needed to quantify a difference of 20% in independent samples with alpha = 0.05 and beta = 0.1 is reduced from 205 for nonpatterned hPSC-CMs to 23 using 2DMBs.

In addition, we found that 2DMBs demonstrated less variability than standard hPSC-CMs in calcium transients in

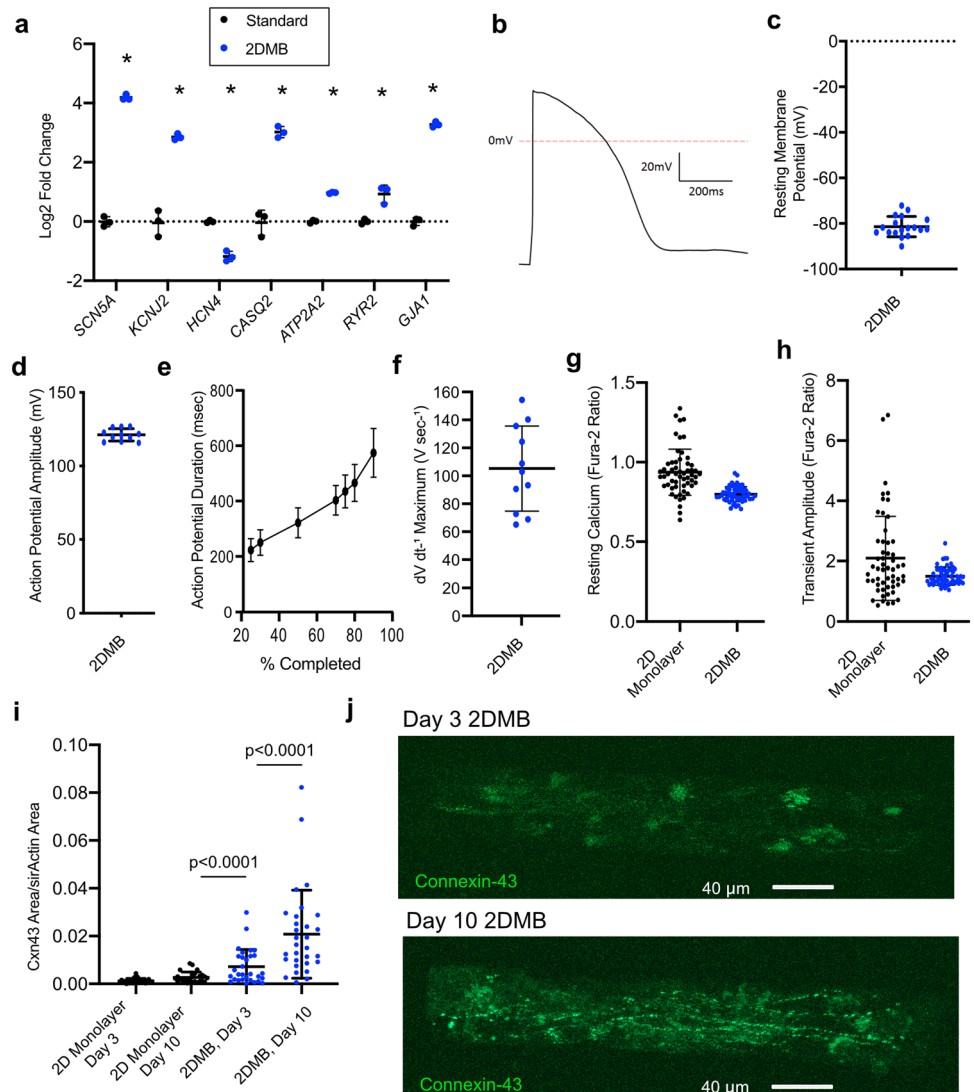

**Fig. 3 Maturation of calcium handling, electrophysiology, and gap junction development is dependent on the biomechanical and contractile environment. a** RNA expression of ion channel and calcium handling genes by RNA-seq in micron-scale two-dimensional cardiac muscle bundles (2DMBs) was compared to standard hPSC-CMs using normalized RNA-seq reads (data points from three biologic replicates per condition from five pooled batches of iPSC-CMs, two-sided $P$ value shown from unpaired $t$ test; mean ± standard deviation shown by error bars). **b** Representative action potential from an 2DMB by patch clamping. **c–f** Patch clamping of 2DMBs showed reproducible resting membrane potentials (**c**), action potential amplitude (**d**), action potential duration (**e**), and maximum dV/dt during phase 0 depolarization (**f**). Resting membrane potentials were measured from 18 2DMBs from five batches and triggered action potentials were recorded for 11 2DMBs from four batches (mean ± standard deviation shown by error bars except in (**e**) where the standard error of the mean is shown). **g** Resting fura-2 ratios were lower in 2DMBs with reduced variability across samples ($N = 56$ from three batches for each condition; two-sided $P$ value shown from unpaired $t$ test; mean ± standard deviation shown by error bars). **h** Calcium-transient amplitudes exhibited less variability in 2DMBs ($N = 56$ from three batches for each condition; two-sided $P$ value shown from unpaired $t$ test; mean ± standard deviation shown by error bars). **i** Connexin-43 quantification by time-lapse immunofluorescence (days 3 and 10) using a live-cell GFP-reporter line (Allen Institute AICS-0053 cl.16) demonstrated greater abundance in 2DMBs with a marked increase between days 3 and 10 ($N = 25$ standard monolayers and $N = 30$ 2DMBs; two-sided $P$ value shown from unpaired $t$ test for 2DMB vs standard monolayer and paired $t$ test for consecutive time points; mean ± standard deviation shown by error bars). **j** Paired images of a representative 2DMB at days 3 and 10 from the connexin-43-GFP-reporter line. Source data are provided as a Source Data file.

response to incremental concentrations of verapamil (Fig. 4g, h). Based on the observed standard deviations, a power analysis revealed that the number of samples needed to quantify a difference of 20% in independent samples with alpha = 0.05 and beta = 0.1 is reduced from 148 to 12 using 2DMBs.

**Physiologic biomechanics drive transcriptional development in hPSC-CMs.** Having shown reproducible contractile behavior and electrophysiologic maturation in 2DMBs, we next used RNA-seq

to quantify the developmental transcriptional state attained in the physiologic biomechanical environment of 2DMBs. We compared 2DMBs to standard 2D monolayers and a variety of biochemical techniques that have been previously shown to enhance maturity in 2D monolayers: (1) oxidative phosphorylation promoting media ("OxPhos", modified from Correia et al.[21]), (2) tri-iodothyronine thyroid hormone (T3)[22–24], (3) T3 plus dexamethasone[23,24], and (4) cell replication inhibition by torin1[25]. Sample groups clustered separately in a principal component analysis but with similar expression across housekeeping genes[26],

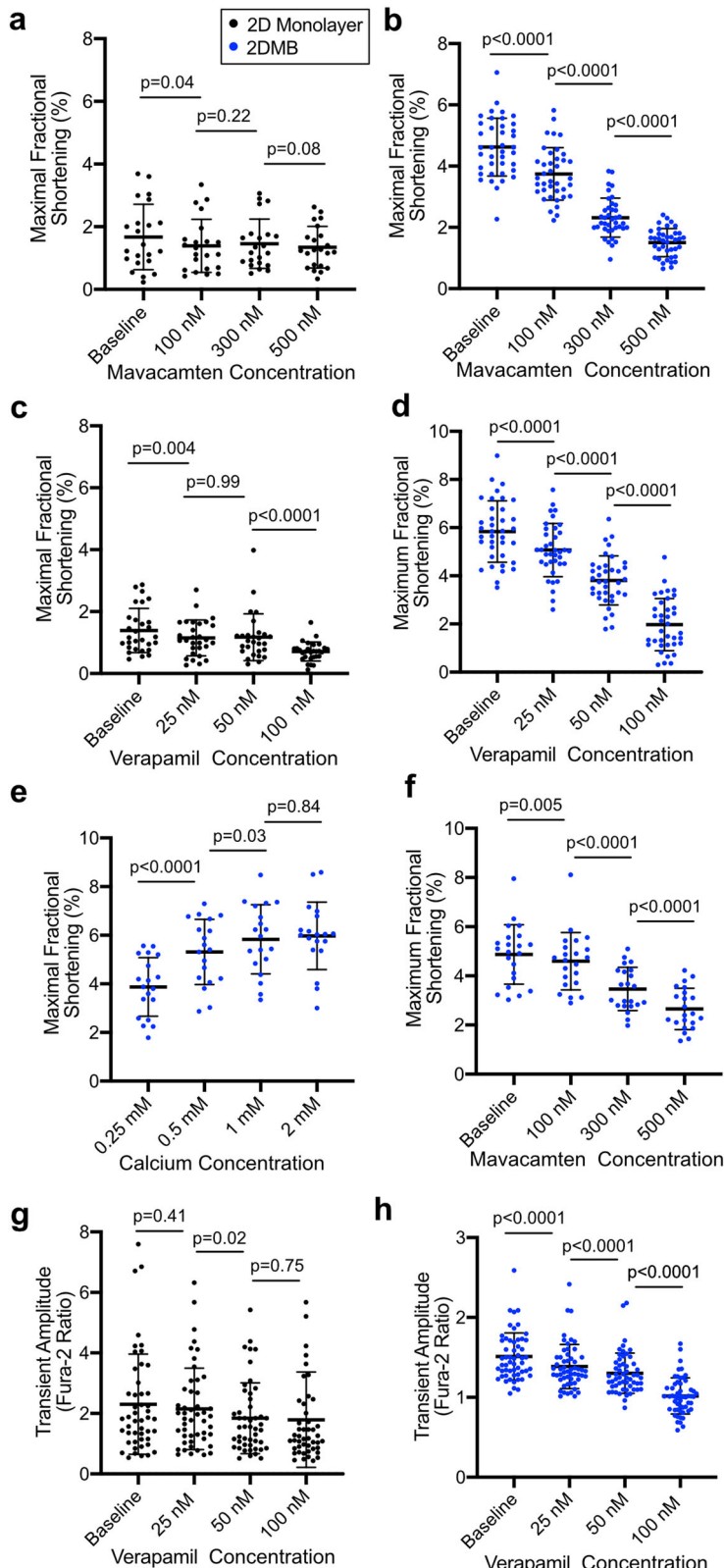

**Fig. 4 Micron-scale two-dimensional cardiac muscle bundles (2DMBs) increase the precision of pharmacologic testing in hPSC-CMs. a**, **b** Maximal fractional shortening with mavacamten in standard hPSC-CMs (**a**) and 2DMBs (**b**) on 8 kPa PDMS (N = 22 and N = 39, respectively). **c**, **d** Maximal fractional shortening with verapamil in standard hPSC-CMs (**c**) and 2DMBs (**d**) on 8 kPa PDMS (N = 29 and N = 37, respectively). **e** Concentration-response of increasing calcium concentration on maximal fractional shortening in 2DMBs (N = 19). **f** Maximal fractional shortening with mavacamten in 2DMBs on 8 kPa PDMS from a different iPSC line (Allen Institute AICS-0053 cl.16, N = 22). **g**, **h** Calcium-transient amplitudes with verapamil in standard hPSC-CMs (**g**, N = 48) and 2DMBs (**h**, N = 55). P values are shown for paired t tests for consecutive time points for all experiments. The mean ± standard deviation is shown by error bars for all experiments. Source data are provided as a Source Data file.

indicating robust normalization with the analysis pipeline (Supplemental Fig. 7A, B). A panel of genes strongly associated with maturation in hPSC-CMs[5] revealed improved maturation with all of the tested strategies compared to control 2D monolayers. However, 2DMBs resulted in the largest incremental gain for all genes except *TNNI3* and *ADRB1* (Fig. 5a, see next section for further studies related to *TNNI3* and *ADRB1*). Using all differentially expressed genes (DEGs, adjusted *P* value < 0.001), we then performed gene ontology (GO) analysis (Supplementary Data File 1). In 2DMBs, GO showed over-enrichment of 561 biologic process categories. Many GO categories across maturation conditions contained overlapping genes and could be summarized in five general types—tissue connectivity, cardiac development, electrophysiology, mitochondrial respiration, and cell proliferation (Fig. 5b and Supplementary Data File 2). All categories exhibited a larger extent of differential expression in 2DMBs, except for mitochondrial respiration. The latter category was primarily differentially regulated in the OxPhos +/− T3 conditions, indicating that mitochondrial development is primarily influenced by the biochemistry of the culture condition distinct from the contractile mechanical cues. KEGG pathway analysis revealed downregulation in DNA replication and cell cycle pathways as among the most statistically significantly differentially regulated pathways in 2DMBs ($P = 1.9 \times 10^{-9}$ and $1.2 \times 10^{-5}$, Supplemental Fig. 8A). Notably, directional changes in cell-proliferation gene expression were similar across other monolayer maturation strategies, but the magnitude of change across these genes was significantly larger in 2DMBS, indicating a marked downregulation of the proliferative transcriptional program in 2DMBs (Fig. 5d). This loss of proliferative potential is consistent with a further shift from the cardiac progenitor state in 2DMBs.

To further identify where on the developmental spectrum 2DMBs reside, we compared genes associated with maturation across 2D monolayers, 2DMBs, 3D hPSC-CM cardiac tissues, and the adult human heart. For the latter two groups, we utilized RNA-seq data that were generated using a similar analysis pipeline by Mills, et al., who created an advanced 3D human cardiac organoid tissue platform in which tissues contract between elastomer posts and are comprised of ~25% fibroblasts[27]. 2DMBs exhibited a similar gene expression pattern for key cardiac genes with directionality toward an adult heart profile (e.g., similarly upregulated *SCN5A*, *KCNJ2*, *MYH7*; downregulated *HCN4*; Fig. 5e). Evidence of incremental myofilament isoform maturation was present in 3D cardiac organoids (i.e., *MYL2*, *TNNI3*), though not attaining the full adult heart gene expression profile, as was previously reported[27]. Recently, *HOPX* was shown to be an important regulator of growth and maturation in hPSC-CMs but was stalled developmentally in standard 2D monolayer hPSC-CMs[28]. Remarkably, we observed a 12-fold increase in *HOPX* in 2DMBs ($P$-adjusted = $1.2 \times 10^{-19}$) and a marked downregulation of *HOPX*'s direct downstream repression targets *AXIN2* (sixfold down, $P = 2.5 \times 10^{-5}$) and *WNT2* (118-fold down, $P = 5.6 \times 10^{-71}$) that are effectors of the proliferative state in cardiac progenitors (Fig. 5f)[29,30]. Also marking a developmental shift from the progenitor state, *TBX2* was downregulated in 2DMBs (4.3-fold down, $P$-adjusted = $3.2 \times 10^{-27}$). This *HOPX-AXIN2-WNT2-TBX2* pattern was similarly observed in 3D cardiac organoids and adult human hearts (Fig. 5f). Taken together, these data indicate that the contractile environment, whether in 2D or 3D, is sufficient to progress cardiomyocyte development from a proliferative-potential state toward a terminally differentiated state.

**Metabolic and hormonal inputs are orthogonal maturation signals in hPSC-CMs**. In the RNA-seq analysis, we found that

two important metrics of cardiac maturation were apparently independent of the contractile environment: *TNNI3* (cardiac troponin I, cTnI) exhibited the largest incremental gain in expression under the OxPhos media condition, and *ADRB1* (beta 1 adrenergic receptor) had the largest incremental gain with T3 hormone (Fig. 5a). Therefore, we performed additional studies to validate these findings, particularly since the isoform switch from *TNNI1* (slow skeletal troponin I, ssTnI) to *TNNI3* is considered an important maturation readout[5,31]. To rigorously validate the RNA-seq findings and investigate the temporal association with the switch to OxPhos media, we performed mass spectrometry with relative isoform quantification in 2D monolayers, finding that OxPhos media induces an immediate increase in cTnI in monolayer hPSC-CMs, attaining 56 ± 4% of total TnI after only 5 days, at only day 22 post-differentiation (Fig. 6a). Further, through qRT-PCR analysis in 2D monolayers, we demonstrate that this switch occurs through differential expression of *TNNI3* mRNA (encoding cTnI) relative to *TNNI1* mRNA (encoding ssTnI), with further augmentation of *TNNI3* by addition of T3 (Fig. 6b).

Next, we compared 2DMBs under either the control media condition or OxPhosT3. The expression of cTnI in 2DMBs in OxPhosT3 was much greater using quantitative immunofluorescence (3608 ± 1801 vs. 1044 ± 626 AUs, $P < 0.0001$, Fig. 6c, d). We also measured contractile function in 2DMBs in each condition. We found that average baseline fractional shortening was lower in OxPhosT3 2DMBs (4.3 ± 1.4% vs. 5.9 ± 2.1%, $P < 0.0001$, Fig. 6e). However, contraction frequency was also lower in OxPhosT3 2DMBs (0.31 ± 0.11 Hz vs. 0.38 ± 0.15 Hz, $P = 0.0001$, Fig. 6f), and, consistent with greater beta-adrenergic receptor expression in OxPhosT3, isoproterenol induced a much greater frequency response (2.8 ± 1.3-fold vs. 1.2 ± 0.2-fold, $P < 0.0001$, Fig. 6g). This greater frequency response was associated with a positive response of contractile velocity to isoproterenol in 2DMBs in OxPhosT3 only (20 ± 0.2% increase, $P < 0.0001$, Fig. 6h). Related to myofilament isoform expression, we also quantified differential splicing associated with maturation in *TNNT2* as an additional marker of myofilament maturation. We found that the adult *TNNT2* transcript variant accounted for 67.4 ± 0.1% of *TNNT2* transcripts in monolayers, 82.6 ± 1.0% of *TNNT2* transcripts in 2DMBs ($P < 0.0001$ vs. control), and 89.9 ± 0.3% of *TNNT2* transcripts in OxPhosT3 2DMBs ($P < 0.0001$ vs. 2DMBs, Supplemental Fig. 9). Taken together, these results indicate that exogenous metabolic and hormonal signals are orthogonal cues further driving distinct developmental endpoints of the myofilament gene program and adrenergic signaling.

**Contractile force and calcium are regulators of hPSC-CM development**. Since the developing heart in vivo is subjected to a hemodynamic pulsatile workload, we hypothesized that contractile force in geometrically organized muscle fibers is a likely contributor to cardiac development through a feedforward mechanism that might explain the observed developmental improvements in 2DMBs. To test whether myofilament growth in 2DMBs is dependent on contractile force itself, we compared 2DMBs cultured in the presence or absence of the cardiac sarcomere-specific contractile antagonist, mavacamten, for 8 days. The higher contractile force in control vs. mavacamten-treated 2DMBs resulted in markedly greater *HOPX*, *MYH7*, and *MYL2* expression, consistent with myofilament growth (Fig. 7a). Interestingly, mavacamten did not affect gene expression associated with electrophysiologic development (*SCN5A*, *KCNJ2*, *GJA1*) in 2DMBs, indicating that these parameters are influenced by different inputs (Fig. 7a). Consistently, the ventricular myosin regulatory light chain (MLC-2v, expressed by *MYL2*), which is a

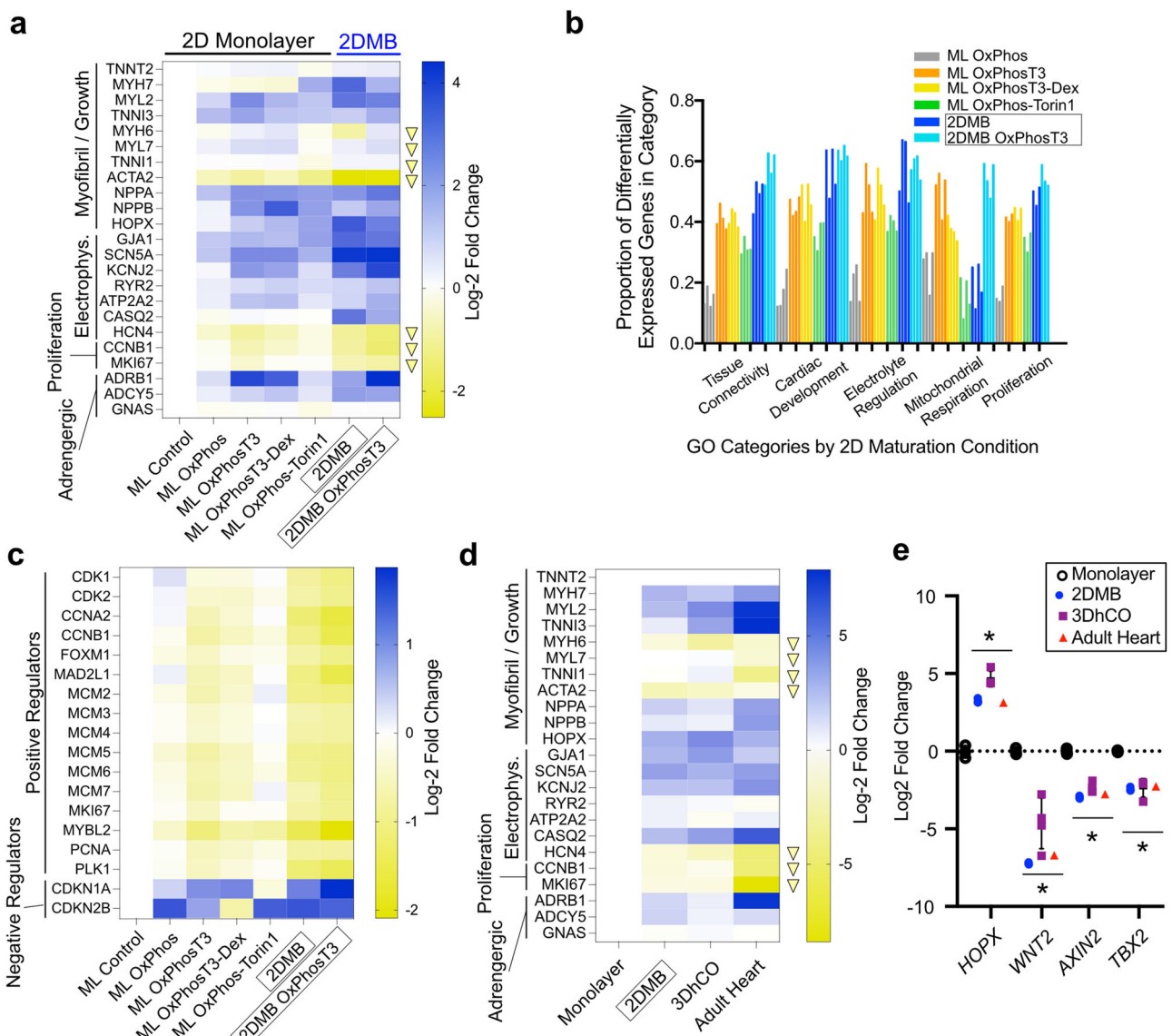

**Fig. 5 Physiologic biomechanics drive transcriptional development in hPSC-CMs. a** Relative expression (log-2 fold change) of genes associated with maturation is shown by heatmap across 2D maturation conditions. Genes expected to be downregulated during maturation are indicated with a downward yellow arrowhead at the right of each row. **b** Gene ontology analysis of maturation conditions versus standard monolayer hPSC-CMs demonstrated overrepresentation of differentially expressed genes among several summative categories: (1) tissue connectivity (e.g., multicellular organism development, extracellular matrix organization, response to mechanical stimulus, cell–cell junction organization shown left–right), (2) cardiac development (e.g., cardiac muscle cell differentiation, actin cytoskeleton organization, (3) cardiac muscle hypertrophy, cardiac ventricle development shown left–right), (4) electrophysiology (e.g., regulation of membrane potential, cardiac muscle cell action potential, membrane repolarization, cellular calcium ion homeostasis shown left–right), (5) mitochondrial respiration (e.g., cellular respiration, oxidative phosphorylation, mitochondrion organization, respiratory electron transport chain shown left–right), and (6) cell proliferation (e.g., regulation of cell migration, regulation of cell proliferation, muscle cell proliferation shown left–right). Data are shown as proportion of DEGs ($P$-adjusted < 0.001) out of all genes in each category (see Supplementary Data File 2). **c** Relative expression (log-2 fold change) of genes associated with cell proliferation is shown by heatmap across maturation conditions. These genes were obtained from DNA replication and cell cycle Kegg pathways as shown in Supplemental Fig. 8. **d** Relative expression (log-2 fold change) of genes associated with maturation is shown by heatmap for micron-scale two-dimensional cardiac muscle bundles (2DMBs) compared to 3D human cardiac organoid tissues (3DhCO) and adult human heart from Mills et al.[27] RNA-seq reads were normalized to *TNNT2* to enable comparison of relative cardiac gene expression independent from other admixed-cell types in 3DhCOs and adult heart. **e** Relative expression of genes associated with the transition from the cardiac progenitor state toward terminal cardiomyocyte differentiation are shown (*$P$ < 0.01 for two-sided $t$ tests 2DMB or 3DhCO vs. monolayers for each gene; specific $P$ values shown in Source Data file). The mean ± standard deviation is shown by error bars. Source data are provided as a Source Data file.

marker both of ventricular lineage specification and maturation, was expressed with greater abundance in control vs. mavacamten-treated 2DMBs (Fig. 7b, c). Relatedly, we hypothesized that calcium handling development in hPSC-CMs is dependent on the local calcium load and might be stymied by the low-calcium

(0.4 mM) RPMI media conditions used in many hPSC-CM protocols. To test the importance of calcium concentration on calcium handling and separate the potential effect from the contractile force, we utilized a SERCA2-GFP-reporter iPSC line and again tested conditions in the presence or absence of

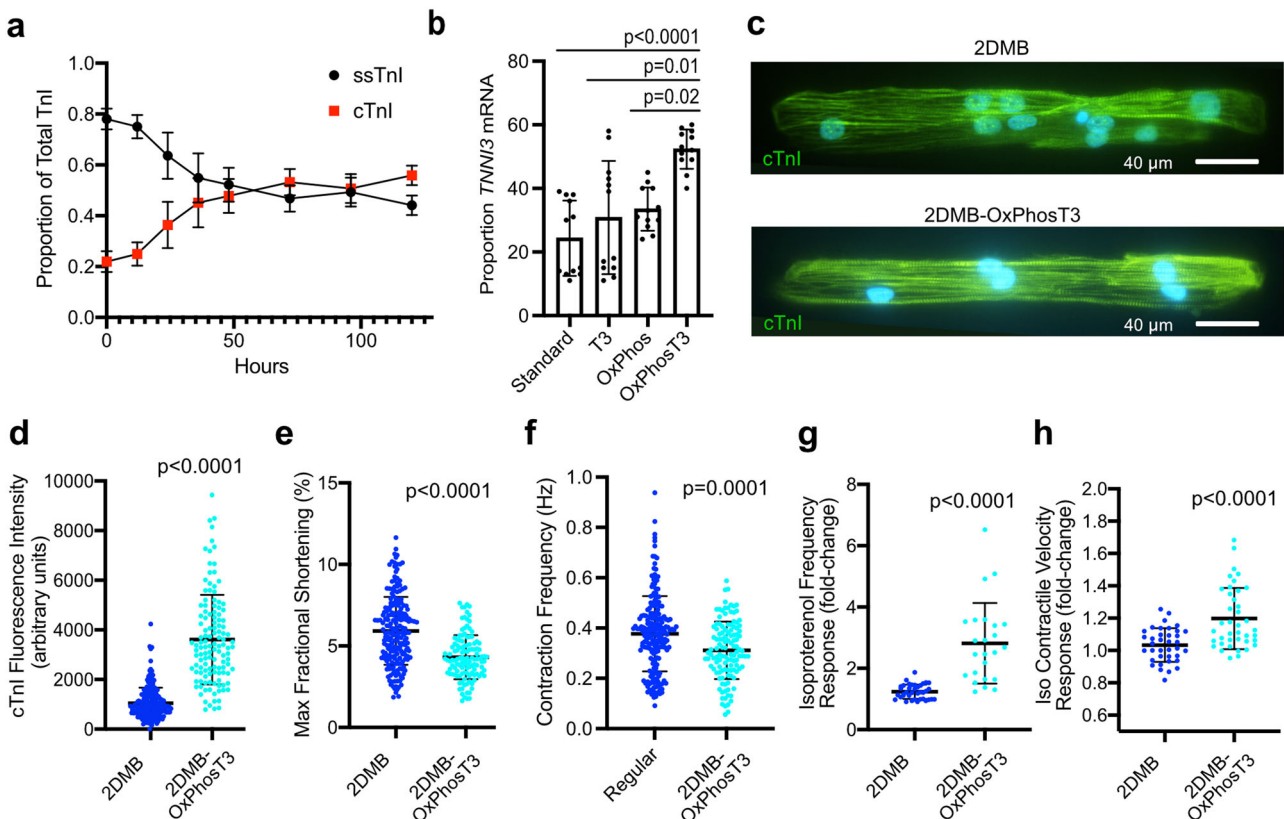

**Fig. 6 Metabolic and hormonal inputs increase myofilament maturation and adrenergic response in micron-scale 2D cardiac muscle bundles (2DMBs) distinct from the biomechanical environment. a** In standard hPSC-CM monolayers, switching to a media that requires oxidative phosphorylation (OxPhos) induces expression of cTnI compared to ssTnI (measured by mass spectrometry, each data point indicates the mean of $N = 6$ biologic replicates at each time point, two each for three different hPSC-CM lines) beginning in the first 24 h after media change (mean ± standard deviation shown by error bars). **b** Relative *TNNI3* mRNA expression (as a proportion to total of *TNNI3* + *TNNI1* mRNA) is increased in standard hPSC-CMs in OxPhos media synergistically with T3 after 5 days ($N = 6$, triplicates for two different hPSC-CM lines). **c** Representative images of cTnI expression in 2DMBs in regular (top) and OxPhosT3 media (bottom). More hPSC-CMs in regular media exhibited minimal myofibrillar expression of cTnI (lower cells in top image). **d** Quantitative immunofluorescence of cTnI in hPSC-CMs in 2DMBs in regular glucose-containing media or OxPhosT3 media ($N = 197$ and $N = 125$, respectively). **e** Contractile quantification in 2DMBs in standard and OxPhosT3 media ($N = 208$ and $N = 122$, respectively). **f** Spontaneous contraction frequency in 2DMBs in standard and OxPhosT3 media ($N = 203$ and $N = 120$, respectively). **g** Isoproterenol frequency response (10 nM) was quantified as fold change from baseline for each 2DMB ($N = 38$ and $N = 24$, respectively). **h** Response of contraction velocity to isoproterenol 10 nM was quantified as fold change from baseline for each 2DMB ($N = 37$ and $N = 40$, respectively). The mean ± standard deviation is shown by error bars for all graphs. Source data are provided as a Source Data file.

mavacamten. Alpha-actinin was used to assess myofibrils and to quantify co-localization of SERCA2 at z-disks, where sarcoplasmic reticula development occurs. Consistent with the above results that demonstrated myofilament growth in response to contractile force, we found that the physiologic-range calcium (1.45 mM) condition we used for 2DMBs resulted in greater alpha-actinin abundance than either 0.4 mM calcium or 1.45 mM calcium plus mavacamten (Fig. 7d). In contrast, SERCA2 abundance (localized at z-disks, Supplemental Fig. 10) was markedly increased in the presence physiologic calcium but was only partially blunted by mavacamten (Fig. 7e, f). In summary, these data show that contractile force development and physiologic-range calcium concentration are critical and separate mechanistic inputs for myofilament and calcium handling development.

**2DMBs improve the precision of contractile kinetic investigation for cardiomyopathy disease modeling.** Finally, we tested whether the biomechanical environment of 2DMBs would allow a facile study of an hPSC-CM cardiomyopathy disease model. We compared control 2DMBs to 2DMBs generated from an isogenic

line in which gene editing was performed to delete the *MYBPC3* promoter (*MYBPC3*[pr−/−])[14]. We previously showed that this line does not express MyBP-C (encoded by *MYBPC3*), but had not quantified contractile kinetics[14]. A complete lack of MyBP-C, as essentially occurs with homozygous truncating *MYBPC3* mutations[14], leads to early-onset, very severe hypertrophic and dilated cardiomyopathy[32]. *MYBPC3*[pr−/−] 2DMBs exhibited similar structural myofibrillar development as isogenic controls (Fig. 8a). Maximal fractional shortening was reduced in *MYBPC3*[pr−/−] 2DMBs ($4.6 ± 1.6\%$ vs. $5.9 ± 2.1\%$, $P < 0.0001$). Normalized contractile velocities were higher during each phase of contraction with a greater magnitude of differences in the mid and late contraction phases in *MYBPC3*[pr−/−] 2DMBs (Fig. 8c–e, j and Supplemental Fig. 11). A more rapid contractile deceleration time was also observed in *MYBPC3*[pr−/−] 2DMBs (Fig. 8f). In contrast, normalized relaxation velocities were lower during each phase of relaxation in *MYBPC3*[pr−/−] 2DMBs (Fig. 8g–i, j and Supplemental Fig. 11). These findings are consistent with the role of MyBP-C presenting a load on, and slowing myofilament velocity in the c-zone of sarcomeres[33], such that ablation of MyBP-C results in increased contractile velocity but reduced relaxation velocity.

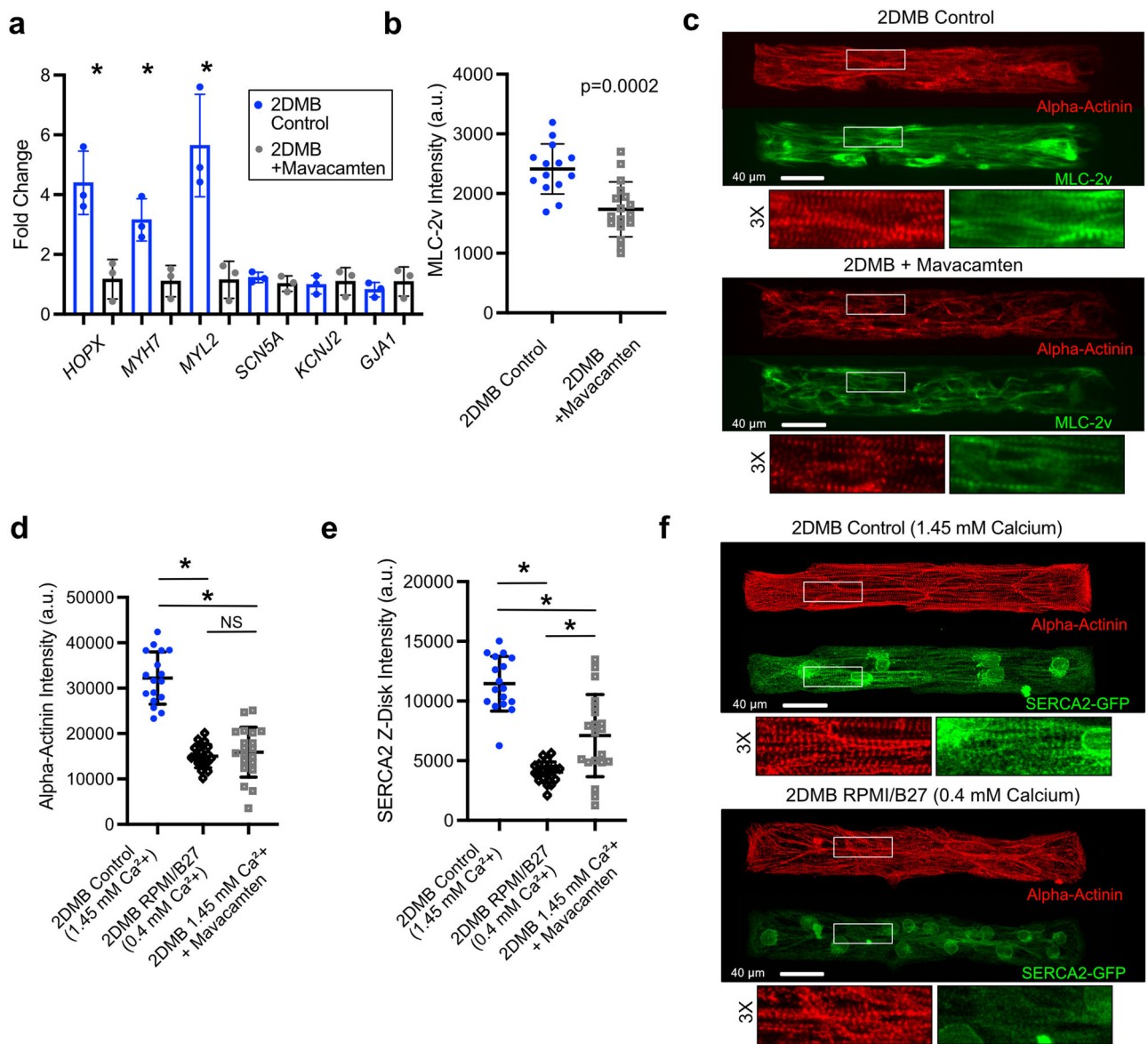

**Fig. 7 Contraction force and calcium concentration are critical for myofilament and calcium handling development in 2DMBs. a** qRT-PCR analysis of micron-scale two-dimensional cardiac muscle bundles (2DMBs) cultured for 8 days with or without the cardiac myosin-specific contractile antagonist, mavacamten (500 nM), demonstrates greater myofilament growth-related gene expression in the control condition. In contrast, genes associated with electrophysiologic maturation are unchanged. Each biologic replicate was obtained from 2 2DMB substrates comprised of >8000 total 2DMBs (DF19-9-11 line). *$P < 0.05$ for two-sided $t$ test (HOPX $P = 0.01$, MYH7 $P = 0.02$, MYL2 $P = 0.01$, SCN5A $P = 0.32$, KCNJ2 $P = 0.75$, GJA1 $P = 0.44$). **b** Immunofluorescence quantification of MLC-2v confirmed increased protein abundance in the absence of mavacamten ($N = 14$ control; $N = 17$ mavacamten). **c** Representative immunofluorescence images of 2DMBs cultured with and without mavacamten for 8 days stained for alpha-actinin and MLC-2v (DF19-9-11 line). **d** 2DMBs cultured for 8 days in near-physiologic calcium (1.45 mM, $N = 17$) exhibited higher alpha-actinin abundance by immunofluorescence quantification than either 2DMBs in standard RPMI/B27 media (0.4 mM calcium, $N = 20$, Tukey-adjusted $P < 0.0001$) or with 1.45 mM calcium media in addition to 500 nM mavacamten ($N = 20$, Tukey-adjusted $P < 0.0001$). 2DMBs were generated from the SERCA2-GFP line. **e** In the same 2DMBs as in **d**, SERCA2 localized to z-disks was increased in the presence of 1.45 mM vs. 0.4 mM calcium ($N = 17$, $N = 20$, Tukey-adjusted $P < 0.0001$). This increase was also observed in the presence of mavacamten with 1.45 mM calcium ($N = 20$, Tukey-adjusted $P < 0.0001$) but was reduced compared to 1.45 mM calcium without mavacamten (Tukey-adjusted $P = 0.0006$). SERCA2 was quantified at z-disks using a co-localization mask with alpha-actinin (see Supplemental Fig. 10). **f** Representative immunofluorescence images of 2DMBs cultured in 1.45 mM and 0.4 mM calcium conditions for 8 days stained for alpha-actinin and anti-GFP (to amplify SERCA2-GFP signal). For all experiments, 2DMBs were first selected for analysis based on brightfield images as described in "Methods", prior to immunofluorescence imaging, to blind 2DMB selection to experimental conditions. The mean ± standard deviation is shown by error bars for all graphs. *$P < 0.05$. Source data are provided as a Source Data file.

## Discussion

Developmental immaturity and heterogeneity, as well as technical hurdles in measuring contractile function, have been major barriers to the reproducible study of cardiac muscle derived from hPSCs[4,5]. Although 3D constructs improve cell organization and maturation, the technical expertise and costs required to generate state-of-the-art 3D tissues are not accessible for many labs, and the complex 3D tissue environment may also limit attempts to efficiently parse the relative importance of mechanical and biochemical cues among mixed cell types. We created a simplified

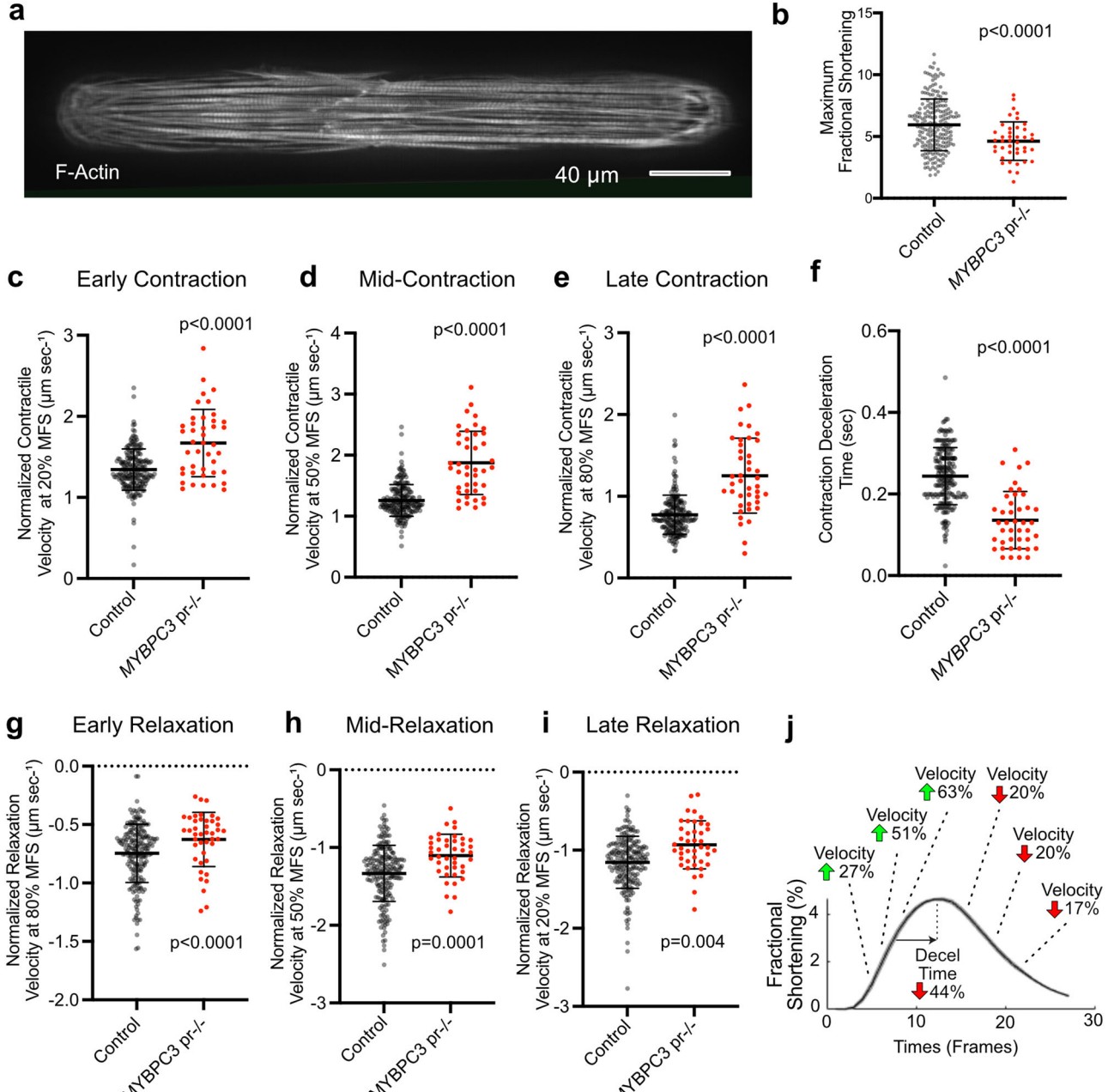

**Fig. 8 Controlling the biomechanical environment improves the precision of contractile kinetic investigation for cardiomyopathy disease modeling. a** Representative example of a cardiomyopathy 2DMB ($MYBPC3^{pr-/-}$) showing normally aligned myofibrils (labeled with SiR-actin). **b** Maximal fractional shortening in control vs. $MYBPC3^{pr-/-}$ 2DMBs ($N = 209$ and $N = 42$). **c**–**e** Normalized contraction velocity in control vs. $MYBPC3^{pr-/-}$ 2DMBs at early (20% of max shortening), mid (50% of max shortening), and late (80% of max shortening) contraction. **f** Contraction deceleration time in control vs. $MYBPC3^{pr-/-}$ 2DMBs (defined as time from deviation from linear contractile slope to peak contraction). **g**–**i** Normalized relaxation velocity in control vs. $MYBPC3^{pr-/-}$ 2DMBs at early (80% of max shortening), mid (50% of max shortening), and late (20% of max shortening) relaxation. **j** Summary of relative differences in contraction and relaxation velocities in $MYBPC3^{pr-/-}$ 2DMBs shown on a representative fractional shortening curve from an $MYBPC3^{pr-/-}$ 2DMB. The mean ± standard deviation is shown by error bars and statistical significance tested using two-sided *t* tests for all graphs. Source data are provided as a Source Data file.

2D approach that isolates biomechanical inputs on cardiac muscle development and demonstrated that the biomechanical environment is independently a critical determinant of myofibrillar and electrophysiologic maturation. Moreover, 2DMBs enable study of integrative effects from biomechanical and orthogonal biochemical signals that affect early cardiac development. Finally, we show that rigorous control of geometry and workload in 2DMBs markedly improves throughput and reproducibility of contractile quantification for pharmacologic testing

and cardiomyopathy disease modeling compared to both 2D monolayers and single micropatterned hPSC-CMs.

Micropatterning or nanogroove techniques have been shown to generate anisotropic 2D cardiac tissues with improved myofibrillar alignment[34,35]. However, generation of myofibrillar alignment without enabling homogeneous auxotonic contractions leads to regional tethering of 2D tissues since the entire substrate cannot be deformed commensurate with fractional shortening (one notable exception being anisotropic thin muscle films)[36].

The key insight enabling our approach was that individual cardiomyocyte assemblies must be uncoupled both from each other and the rigid glass underlying the elastomer surface to enable auxotonic contractions unperturbed by neighboring cells. We designed 2DMBs to control hPSC-CM force transmission by scaling micropatterns to a size that enables (1) hPSC-CM geometric organization, (2) separation of individual 2DMBs to avoid physical interactions, and (3) live-cell quantification. This approach generated arrays of individual muscle bundles that rapidly align myofibrils, develop symmetric and homogeneous contractile behavior, and exhibit improved electrophysiology. Comparisons to standard 2D plating conditions demonstrate that myofibrillar alignment and auxotonic contractions in 2DMBs are sufficient to drive a significant level of hPSC-CM maturation even without the complex, admixed-cell environment of 3D collagen- or fibrin-based tissues. Compatibility with the large and growing library of Allen Institute GFP-reporter hPSC lines also enables efficient analysis of developmental cues affecting key structural proteins, as we showed with connexin-43 and SERCA2.

A major motivation for this study was the variability in contractile phenotypes and the extensive time required to perform contractile analyses using the traction force method in single micropatterned hPSC-CMs[14,37]. Multicellular connectivity and uniaxial contractions of 2DMBs were associated with improved myofibrillar development, resulting in far greater throughput and reproducibility. The efficiency of contractile analysis was further increased by the ContractQuant algorithm, which reduced the computational time by ~900-fold compared to bead-tracking with traction force microscopy. Moreover, brightfield imaging avoids fluorescence-induced phototoxicity as a potential experimental confounder. Using 2DMBs with ContractQuant analysis allowed precise characterization of a cardiomyopathy hPSC-CM model with contractile kinetic dysregulation due to *MYBPC3* ablation that reinforces key findings from biophysical studies[33]. As compared to 3D microtissues or traction force microscopy of single hPSC-CMs, both the platform and analysis approaches developed here could be readily and cost-effectively implemented in most laboratories with hPSC-CM experience.

In addition, 2DMBs enable testing of early cardiac development in a reductionist and controlled setting of purified and mechanically organized hPSC-CMs. Our analysis showed evidence of maturation with several previously published techniques in standard 2D monolayers, but a larger gain was attained with 2DMBs, emphasizing the importance of a physiologic contractile environment for cardiomyocyte development toward a terminally differentiated state (as marked by increased *HOPX*, reduced *WNT2*, *TBX2*, and cell-proliferation genes). We further showed that contractile force itself is a critical input for early-stage myofilament development and that a physiologic calcium concentration is sufficient to drive marked improvements in sarcoplasmic reticula development with proper z-disk localization (as marked by SERCA2). Superimposed on the biophysical environment, induction of oxidative phosphorylation in combination with thyroid hormone synergistically increased cTnI expression, the mature *TNNT2* transcript variant, and the beta-adrenergic response. This analysis demonstrates the utility 2DMBs to examine combinatorial influences of early cardiac developmental signals on specific endpoints in a controlled contractile environment.

The 2DMB platform has limitations. The improved reproducibility of the contractile environment of 2DMBs does not remove the innate heterogeneity in differentiation efficiency and cell-type specification that inevitably occurs across different hPSC lines and batches with current methods. However, this reductionist platform may be useful, particularly with its compatibility with 2D image-based quantification of reporter lines, to better understand how differentiation techniques affect cell fate and development in a mechanically controlled environment. In addition, while the intentional lack of stromal cells in 2DMBs is a potential advantage to simplify some mechanistic and disease modeling applications, the lack of other cell types may also limit some aspects of cardiomyocyte development important for other applications. Finally, although adherence of 2DMBs to micropatterned PDMS was excellent for the experiment durations reported here, we observed that prolonged culture times in a high (physiologic) calcium environment were not sustainable since the associated progressive increase in contractile force of 2DMBs eventually overcomes focal adhesion strength on micropatterned PDMS. Thus, experiments requiring chronic culture conditions in an organized contractile environment would require either fibroblast-enriched 3D tissues or more strongly supportive 2D extracellular matrix scaffolds[38].

In conclusion, our results show that the biomechanical environment of cardiac muscle cells derived from hPSCs is a critical determinant of their early developmental maturation and contractile function. The controlled geometry and elastic workload of 2DMBs markedly improve myofibrillar and electrophysiologic function, enabling reproducible contractile quantification. 2DMBs can also serve a critical role as a test bed to study early cardiac development and disease in a biomechanically controlled and simplified environment, enabling subsequent scale-up to more complex 3D systems.

## Methods

**Stem cell and cardiomyocyte culture**. The use of iPSCs for this project was approved by the University of Michigan Human Pluripotent Stem Cell Research Oversight Committee. Control iPSCs were obtained from WiCell (iPS-DF19-9-11T), Coriell (WTC-11, GM25256), and the NYSCF Research Institute Stem Cell Repository (MR18, MR30, MR48). GFP-labeled iPSCs were obtained from the Allen Institute (connexin-43-GFP, AICS-0053 cl.16; SERCA2-GFP, AICS-0046 cl.51; DSP-GFP, AICS-0017 cl.65). Donors of cells previously gave informed consent prior to inclusion in the respective repository above, and their identities are anonymous. IPSCs were cultured in StemFlex (Gibco) and verified free of mycoplasma using MycoAlertTM (Lonza). Cardiac differentiations were performed using Wnt modulation[1,2]. We achieved optimal differentiations using RPMI plus B27 supplement without insulin during day 0–2 (CHIR 5–6 μM during first 24 h LC Laboratories), then CDM3 media (without B27) with IWP4 5 μM (Stemgent) during days 3–4. Switching to CDM3 at day 3 avoids retinoic acid (included in B27), and we additionally included retinol inhibitor (BMS 453, Cayman Chemical, 1 μM) for days 3–6 to minimize atrial lineage differentiation[39]. hPSC-CMs were maintained in CDM3 until purification in CDM3-lactate using glucose-deprived, lactate-containing media on days 12–16 with one modification— we used RPMI without glutamine (Biologic Industries) for more consistent purification[2,40]. Purified hPSC-CMs were replated as monolayers (400,000 cells/cm²) for 8 additional days (until day 24) on stiff PDMS membranes (Specialty Manufacturing, Inc.) with growth factor reduced Matrigel (Corning) in CDM3-lactate media (4 days) and then RPMI plus B27 supplement (4 days)[41]. On day 24, hPSC-CMs were dissociated by 0.25% Trypsin-EDTA (Gibco) with 5% (v/v) Liberase (Sigma-Aldrich) for 5 min, stopped by an equal volume of 20% FBS/1 mM EDTA/PBS (low calcium at this step improves viability). Cells were triturated by gently pipetting with a p1000 pipette eight times to obtain a near single-cell suspension and centrifuged (150 g, 3 min). hPSC-CMs were resuspended in replating media (RPMI plus B27 supplement with 2% FBS), filtered (40-μm strainer), counted (LUNA™, typical viability ~99%), and 45,000 cells were seeded per 2DMB substrate within 1.2 ml in the mini-well of devices. Two hours after cell seeding, the media was gently removed without drying the substrate and 4 ml of fresh replating media was added back in the dish. On the following day, the media was replaced with 4 ml of 50:50 RPMI:DMEM with 1× B27 supplement. The media calcium concentration was gradually raised from 0.4 mM in RPMI to 1.45 mM by replacing half volume in the dish every 2 days with 25:75 RPMI:DMEM with 1× B27. 2DMBs were assessed after 8 additional days (i.e., day 32 post-differentiation) except where indicated. All media were equilibrated (30 min at 37 °C and 5% $CO_2$) before use.

For maturation testing, we utilized a media intended to increase oxidative phosphorylation, referred to as "OxPhos". This media, adapted from Correia et al., is based on glucose-free RPMI with 1× B27 supplement and galactose, lactate, glutamax, and pyruvate at final media concentrations of 4, 4, 2, and 0.5 mM, respectively (all from Sigma)[42]. A low concentration of fatty acids is present in the B27 supplement and additional fatty acids were not added to avoid the potential for oxidation and free radical formation. Triiodo-L-thyronine (T3) was used at 100 nM (Sigma), dexamethasone at 1 μM (Cayman), and torin1 at 200 nM (Cayman).

**Micropatterned substrate fabrication**. To create the 2DMB platform, micro-pattern geometries were first designed in AutoCAD by scaling up our prior single-cell micropatterning approach[14,37]. The design consisted of a rectangular array of 4772 rectangles with a dimension of $308 \times 45 \, \mu m$ (area $13,952 \, \mu m^2$), spaced by $240 \, \mu m$ from each other in the rectangles' long-axis direction and by $80 \, \mu m$ from each other in the rectangles' short axis direction. Standard soft lithography techniques were used to create a silicon master mold with micropattern designs from which polydimethoxysiloxane (PDMS) stamps were made. A 0.09-inch thick, $5 \times 5$-inch chromium photomask containing six replicates of each design was produced by Photo Sciences, Inc. (Torrance, CA, USA) on a soda lime glass substrate at $\pm 1 \, \mu m$ feature tolerance and with the chrome side down. Microfabrication of the silicon master mold followed the standard protocol developed by Microchem, Inc. for SU-8 2005. Briefly, a 100-mm silicon wafer was first dehydrated on a hot plate at 150 °C for 5 min. Then a negative photoresist, SU-8 2005, was spin-coated on the silicon wafer at 500 rpm for 10 s and 1000 rpm for 30 s, which gave a thickness of ~8.6 μm. The SU-8 photoresist was then exposed to UV light at $115 \, mJ/cm^2$ for 9 s under a contact aligner (Karl Suss, MJB45) with the chromium photomask. After the development of the SU-8 photoresist, the silicon wafer was hard-baked at 200 °C for 10 min. The thickness of the photoresist was measured using a stylus profilometer (Dektak 6 M). The silicon wafer was silanized by placing in a vacuum with two drops of trichloro(1H,1H,2H,2H-perfluorooctyl)silane placed on aluminum foil. To prepare the PDMS stamps, Sylgard-184 elastomer and curing agents (Dow Corning, Midland, MI) were mixed at a ratio of 10:1 (w/w), degassed, then cast over the silicon mold and cured at 50 °C overnight. Individual stamps were cut from the cured PDMS. Individual stamps can be re-used innumerable times as long as the surface is not inadvertently scratched.

Initial micropatterning testing was performed on 8.7 kPa polyacrylamide hydrogels functionalized for direct fibronectin stamping by creating a pre-polymer mixture containing the following: 400 μl 40% acrylamide (Sigma), 65 mg of N-hydroxyethyl acrylamide (Sigma) dissolved in 1 ml 50 mM 4-(2-hydroxyethyl)-1-piperazineethanesulfonic acid (HEPES, Sigma), 250 μl 2% bis-acrylamide (Sigma), and 3285 μl HEPES[14,16]. Polymerization was induced with 25 μl of 10% ammonium persulfate (Sigma) and 2.5 μl of tetramethylenediamine (TEMED, Sigma). These functionalized gels were then microprinted with 1:25 diluted 0.1% human serum-derived fibronectin (Sigma) that was first incubated on PDMS stamps as described below. Since this method failed to maintain adherence of 2DMBs, final optimized 2DMB substrates consisted of micropatterned 8 kPa PDMS. Soft PDMS at specified stiffness was formulated by mixing Sylgard® 527 and Sylgard® 184[17]. Specifically, each component was first mixed with its own curing agent (i.e., 50:50 for Sylgard® 527 and 10:1 for Sylgard® 184). To obtain 8 kPa PDMS, a 1:97 ratio (mass:mass) of Sylgard-184: Sylgard 527 was mixed thoroughly, then degassed.

Glass bottom dishes were fabricated by first gluing a 40-mm diameter glass coverslip underneath a 6-cm culture dish with a 25-mm hole drilled through the bottom using silicon all-purpose adhesive sealant (DAP), creating a 25-mm mini-well in the center of the 6-cm dish. In all, 135 μl of the PDMS mixture above was pipetted onto the center of each mini-well with a P1000 pipette and low-viscosity tip to create a ~70-μm thick layer of PDMS. Since 8 kPa PDMS is too tacky for direct microcontact printing, a two-step transfer process with a fibronectin micropatterned polyvinyl alcohol (PVA) film intermediary was used[18,19]. To create the PVA film, 65 ml of 5% (w/v), 70-μm strainer filtered, PVA solution was poured onto a $20.5 \times 25.5$ cm leveled flat glass panel and left to air dry (2–3 days). Completely dried PVA films were peeled and trimmed. For print micropatterns, PDMS micropatterning stamps were first cleaned in 50% ethanol sonication bath. 350 μl of 1:25 diluted 0.1% human serum-derived fibronectin (Sigma-Aldrich) was added per stamp and incubated at room temperature for 1 h, then aspirated. Following air-drying, stamps were pressed on pre-cut, individual stamp-size PVA films and incubated for 30 min. The PVA films with fibronectin micropatterns were then peeled and inverted onto cured 8 kPa PDMS. Microprinted dishes were stored up to 2 weeks at 4 °C prior to use with the PVA films remaining adherent. Immediately prior to use, PVA films were dissolved in sterile water at room temperature on an orbital shaker (70 RPM) for 15 min to expose the micropattern, and surfaces were sterilized by UV in the culture hood for 10 min.

**Microscopy**. Microscopy was performed at 37 °C and 5% $CO_2$ for live-cell experiments. DIC microscopy was performed using a Nikon Eclipse Ti-E inverted microscope with motorized stage and an Andor Zyla camera. 2DMBs filling micropatterns with ~6–12 cells per pattern were first identified at ×10 and locations saved to increase the efficiency of imaging. A ×40 air gap objective was then positioned for contractile imaging (avoiding heat sink from oil immersion) and time-series images were obtained for each 2DMB with the field of view $330 \times 100 \, \mu m$ at 50 frames per second. All hPSC-CMs were labeled for F-actin with the cell-permeant dye, SiR-Actin (Cytoskeleton, Inc.) prior to imaging (0.5 μM, 1 h) to enable visualization of myofibrils following contractile imaging, and this protocol enabled confirmation that all selected 2DMBs were 100% pure cardiomyocytes in every experiment. Using the tagged locations of 2DMBs previously imaged for contractions, F-actin images were obtained as a z-stack in fixed (4% paraformaldehyde) 2DMBs with 1-μm slice thickness, and deconvolutions with maximum intensity projections were performed in ImageJ.

Time-lapse imaging of the connexin-43-GFP-reporter line was performed with confocal microscopy at ×40. For the connexin-43 imaging only, mavacamten was first administered at 500 nM prior to imaging to temporarily reduce fractional shortening to minimize blurring during z-stack acquisitions with 0.5-μm slice thickness. Image locations were saved during imaging at day 3, and the same 2DMBs were imaged again at day 10. As above, SiR-Actin staining was performed prior to both day 3 and day 10 for concurrent imaging of myofibrils. Maximum intensity projection images were obtained from each z-stack. Myofibrillar orientation images were obtained by labeling fixed (4% paraformaldehyde) 2DMBs with an MyBP-C antibody (N-terminal rabbit polyclonal, 1:1000, kind gift generated in the lab of Samantha Harris) and goat anti-rabbit AlexaFluor 568 secondary (1:1000, ThermoFisher A-11011). Intercalated disk images were obtained by labeling fixed 2DMBs with an N-cadherin antibody (1:200, BD Biosciences 610921) and goat anti-mouse AlexaFluor 488 secondary (1:1000, ThermoFisher A-11001). Antibodies against alpha-actinin (1:1000, Sigma A7811) and Mlc-2v (Proteintech 10906-1-AP, 1:500) were also used to image sarcomere structures with the same corresponding secondaries as above.

**Image processing**. Contractile function was quantified from DIC images using a pixel cross-correlation method that we implemented in MATLAB, referred to as ContractQuant. ContractQuant tracks displacement of 2DMBs by solving the best match for regions of interest (ROIs) placed at the 50% inner length of each 2DMB. ROIs can be modified or manually selected for different imaging platforms. ContractQuant exports the following measurements: (1) contractile velocities at 20%, 50%, and 80% of peak contraction; (2) relaxation velocities at 20%, 50%, and 80% of peak contraction; (3) max fractional shortening; (4) contraction acceleration; (5) contraction deceleration time; (6) relaxation time. Standard deviations for each parameter and graphs of displacement, velocity, acceleration, and a symmetrical motion index are also exported, enabling rapid confirmation of accurate motion tracking. Rare cases with inconsistent motion tracking were excluded from the analysis.

Individual sarcomere unit orientations were measured across images using a separate automated imaging method in MATLAB that creates a binary mask of segmented sarcomere units. These orientations were fit to a Gaussian distribution for each image to calculate the standard deviation of sarcomere angles per image. Myofibrillar abundance in 2DMBs was measured using a previously described MATLAB script (MyoQuant) that detects myofibrils traversing the long axis of 2DMBs based on F-actin signal and then quantifies the density and heterogeneity of these structures[11].

**Intracellular calcium quantification**. Cardiomyocytes were loaded with 1 μM fura-2 AM (ThermoFisher) for 10 min, rinsed with 250 μM probenecid in HBSS for 5 min, changed to pre-equilibrated media, and then incubated for 30 min prior to imaging through a Nikon Super Fluor ×40 objective. The background-subtracted emitted light intensity following alternating excitation at 340 nm and 380 nm was reported as fura-2 ratios.

**Patch clamping**. Whole-cell patch clamp was performed in current-clamp mode using an Axopatch 700B amplifier and Digidata 1440 A (both from Axon Instruments) at room temperature. 2DMBs were bathed in a solution containing (in mM): 135 NaCl, 4 KCl, 1.8 $CaCl_2$, 1 $MgCl_2$, 10 HEPES, 1.2 $NaH_2PO_4$, 10 glucose, pH 7.35 with NaOH. Patch pipettes (2–3 MΩ) were filled with the internal solution containing (in mM): 130 K-aspartate, 10 KCl, 9 NaCl, 0.33 $MgCl_2$, 5 Mg-ATP, 0.1 GTP, 10 HEPES, 10 glucose, pH 7.2 with KOH. The resting membrane potential (RMP) was determined under the current clamp at zero current. Action potentials (APs) were elicited by 20 pulses of 0.4 nA each at 0.5 Hz. The liquid junction potential between bath solution and pipette solution was calculated with pClamp and corrected after experiments.

**Transcript quantification**. Quantification of mRNA for individual genes was performed using TaqMan™ assays from ThermoFisher (*TNNI3*, Hs00165957_m1; *TNNI1*, Hs00268531_m1; *HOPX*, Hs05028646_s1; *MYL2*, Hs00166405_m1; *MYH7*, Hs01110632_m1; *SCN5A*, Hs00165693_m1; *GJA1*, Hs00748445_s1; *KCNJ2*, Hs00265315_m1). RNA-Seq was performed on five pooled batches of hPSC-CMs with verified >95% cardiomyocyte purity. These lactate-purified batches had previously been cryopreserved at day 16 to enable pooling of batches together in a single thaw for comparison of experimental conditions independent from the potential batch effect. Each condition was performed in triplicate and cultured for 8 days prior to collection. RNA samples were processed in a single session using the RNAeasy kit (Qiagen). All samples used for RNA library preparation had an RNA abundance >25 ng/μl, 260/280 ratio >1.9 (by NanoDrop™), and RNA integrity number >8.0. A stranded and barcoded RNA library was prepared and sequenced using a multiplexed library on an Illumina NovaSeq by the University of Michigan DNA Sequencing Core. Reads were aligned to the genome using STAR[43]. Lowly expressed genes were filtered, leaving a dataset of 20,220 genes. Reads per gene were normalized to the library size for each sample using DESeq2[44]. Relative abundance for each gene was determined using dispersion estimates for each gene fit to a binomial generalized linear model with DESeq2[44]. The Wald test with multiple testing correction for 20,220 genes tested was performed to determine statistical significance for individual genes (adjusted *P* value

<0.001)[44]. GOseq was used to test for enrichment of biologic pathways among differentially regulated genes with multiple testing corrections for all pathways tested[45]. Transcriptomics data were also analyzed and visualized using Advaita Bioinformatic's iPathwayGuide (v2006, https://www.advaitabio.com/ipathwayguide) and the Kyoto encyclopedia of genes and genomes (KEGG)[46,47]. Adjusted *P* value <0.05 and log-2 fold change >±0.01 thresholds were used to designate genes as differentially expressed for pathway analyses. These data were deposited in NCBI's Gene Expression Omnibus (GEO accession GSE183398). RNA-seq data from this study were also compared to cardiac organoid RNA-seq data from Mills et al. (GEO accession GSE93841)[27].

**Mass spectrometry.** The effect of OxPhos media on relative cTni and ssTnI abundance was quantified using mass spectrometry to quantify unique peptides in both cTnI and ssTnI. These data were analyzed from a previously described experiment, in which pooled hPSC-CMs from three separate lines (DF19-9-11 and 2 *MYBPC3* patient lines, kind gift from Christine Mummery) were switched at time 0 to OxPhos media[14]. Cryopreserved iPSC-CMs from three different batches were thawed and pooled together for each of the three iPSC lines at the beginning of the experiment to synchronize all subsequent time points. After combining iPSC-CMs for each line from thawing, iPSC-CMs were resuspended was OxPhos media. The media was subsequently changed every 48 h. Duplicate samples were collected at each of the designated time points for each cell line, rinsed with PBS containing protease inhibitors (Roche), and frozen at −80 °C until processing. Upon thawing, iPSC-CMs were solubilized in 75 µl 0.1% RapiGest SF Surfactant (Waters Corporation) at 50 °C for 1 h. The samples were reduced by adding 5 µl of 0.1 M dithiothreitol and heating (100 °C, 10 min), then alkylated with 10.4 µl of 100 mM iodoacetamide in 50 mM ammonium bicarbonate with incubation in the dark (22 °C, 30 min). The proteins were digested into tryptic peptides with the addition of 25 µl of 0.2 µg/µl trypsin (Promega) and incubating (37 °C, 18 h). The samples were dried down in a speed vacuum device. In total, 100 µl of 7% formic acid in 50 mM ammonium bicarbonate was added to deactivate the trypsin and cleave the RapiGest and heated (37 °C, 1 h). The sample was dried down again. 100 µl of 0.1% trifluoroacetic acid was added and heated (37 °C, 1 hour) to cleave the RapiGest again. The resultant peptides were dried down and reconstituted for a final time in 60 µl of 0.1% trifluoroacetic acid. The tubes were then centrifuged at 14,000 rpms for 5 min to pellet the surfactant. The top 55 µl of the solution was transferred into a mass spectrometry (MS) analysis vial.

Peptides were separated by liquid chromatography and their abundances were measured by mass spectrometry. A 20 µl of each sample was injected onto an Acquity UPLC HSS T3 column (100 Å, 1.8 µm, 1 × 150 mm) (Waters Corporation) attached to an UltiMate 3000 ultra-high pressure liquid chromatography (UHPLC) system (Dionex). The UHPLC effluent was directly infused into a Q Exactive Hybrid Quadrupole-Orbitrap mass spectrometer through an electrospray ionization source (ThermoFisher Scientific). Data were collected in data-dependent MS/MS mode with the top five most abundant ions being selected for fragmentation. The resultant.RAW files containing the mass spectra were processed using the SEQUEST algorithm in the Proteome Discoverer 2.2 (PD 2.2) software package (ThermoFisher Scientific). The MS/MS spectra were matched to theoretical spectra generated using the human proteome database. Variable mass changes accounting for the loss of methionine from the N-terminus of each protein with the addition of acetylation (−89.16 Da), the carbamidomethylation of cysteine (57.02 Da), oxidation of methionine and proline (M, P; 15.99 Da: M; 32.00 Da) and phosphorylation serine, threonine and tyrosine (79.98 Da) were added. The area of the peak for each peptide detected in the MS/MS spectra was determined using PD 2.2. The abundance of cTnI and ssTnI were calculated at each time point using the three most abundant unique peptides from each protein. The relative proportion at each time point was calculated as cTni/(cTnI+ssTnI) and ssTnI/(cTnI+ssTnI), respectively.

**Statistics and reproducibility.** Continuous variables were compared by two-sample *t* tests or ANOVA with multiple testing correction (Tukey test) for normally distributed outcomes. Serial measurements in the same 2DMBs were compared with the paired-samples *t* test. Data were collated in Microsoft Excel. Statistical tests and graphs were generated using GraphPad Prism. RNA-seq statistical analyses are described above. Reproducibility of the 2DMB technique was verified in all iPSC lines tested, including all lines used in this manuscript in at least three differentiation batches per line. All data points indicate biological (not technical) replicates. All quantifications in graphs were performed using automated and/or standardized algorithms blinded to experimental conditions. Findings in representative micrographs (Figs. 1c–e, 2b, 3j, 6c, 7c, 7f, 8a and Supplementary Fig. 1b) were similar across biologic replicates and quantified in the corresponding graphs for each finding, with the number of biologic replicates for each given in the corresponding figure legend.

**Reporting summary.** Further information on research design is available in the Nature Research Reporting Summary linked to this article.

## Data availability
The RNA-seq data in this publication have been deposited in the GEO database under accession code "GSE183398". Re-analysis of previously published data from a mass spectroscopy experiment was performed for Fig. 6a[14]. All other relevant data supporting the key findings of this study are available within the article and its Supplementary Information files or from the corresponding author upon reasonable request. Source data are provided with this paper.

## Code availability
Code for both ContractQuant and MyofiberQuant is available at GitHub (https://github.com/Cardiomyocyte-Imaging-Analysis)[48,49].

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

## Acknowledgements
This work was funded by the NIH (K08HL130455 and R03HL148465 to A.S.H., R21GM134167 to A.P.L., R01HL149363 to L.L.I.), NSF (NSF EEC-1647837 and 2033654 to A.S.H. and B.M.B., CMMI-1561794 to A.P.L.), Michigan Institute for Clinical and Health Research Postdoctoral Translational Science Program fellowship from UL1TR002240 to Y.-T.Z., the University of Michigan Research Stimulus Award (A.S.H. and A.P.L.), and the Frankel Cardiovascular Center McKay Award (A.S.H.).

## Author contributions
Conceptualization—A.S.H. and A.P.L.; methodology—Y.-C.T., Y.-T.Z., A.C., S.J.D., Y.W.W., K.U., M.J.P., B.M.B., D.N., L.L.I., A.P.L., and A.S.H.; investigation—Y.-C.T., Y.-T.Z., A.C., S.J.D., B.E., M.I.P., K.U., D.M., S.F., T.S.O., N.W., and K.H.; writing—Y.-C.T. and A.S.H.; writing and review/editing—Y.-C.T., S.J.D., M.J.P., B.M.B., D.N., L.L.I., A.P.L., and A.S.H.; funding—Y.-T.Z., L.L.I., B.M.B., A.P.L., and A.S.H.; supervision—M.J.P., B.M.B., D.N., L.L.I., A.P.L., and A.S.H.

## Competing interests
The authors declare no competing interests.

## Additional information

**Peer review information** *Nature Communications* thanks Wolfram Zimmermann and the other anonymous reviewer(s) for their contribution to the peer review this work. Peer reviewer reports are available.

