## [Peer Review File · Nature Communications]

Reviewers' Comments:

Reviewer #1:

Remarks to the Author:

Tsan and colleagues present an advanced monolayer cardiomyocyte culture method, comprising ensembles of 6-12 cells on micropatterns. Patterns on soft (8 kPa) fibronectin-coated PDMS substrates with 7:1 (308 μm x 45 μm) dimensions are used instead of the often employed smaller (e.g., 70 μm x 10 μm) single cell sized patterns. The increase in pattern size overcomes the well-known limitations associated with single cardiomyocyte growth. Ensemble cardiomyocyte cultures on large patterns were compared to "standard" monolayer cultures without spatial constraints. The study applies a broad range of methods, including numerical simulations, electrophysiology, RNA-seq, mass spectrometry. While an advance over "standard" monolayer cultures with a well-known poor degree of maturation is plausible and expected, there is no data in support of an advance over single cells on patterns or 3D culture models. The authors should be more specific as to the real advantage of their model, which is very likely not maturity, but could be applicability in high throughput assays with an increase in robustness compared to single cardiomyocytes on patterns.

Main critiques:

All claims as to an improvement over 3D models must be deleted, unless the authors include a state-of-the-art 3D model for direct comparison in their study. The study is essentially on 2D with a nice demonstration that ensembles of cells rather the individual cells may have advantages as to applications in drug testing and disease modelling. It is highly unlikely and not supported by any data, that maturity in 2D cultures, irrespective of their format, would reach closely the degree of maturation that can be achieved in state-of-the-art 3D models. For clarity, all wording suggesting a tissue model must be deleted.

The claim that the model overcomes the use of stromal cells is not used within the right context. In monolayer cultures, such as presented by the authors, non-myocytes have never been a requirement. In fact, in such models, non-myocyte number is always kept low to avoid overgrowth. Even in 3D models, stromal cells are not an absolute requirement for individual cardiomyocyte seeding and function. In 3D models, stromal cells are required to organize a complex cardiomyocyte maturation supporting biological [including, but not limited to fibronectin]/biophysical [8-12 kPa] ECM environment. As such, state-of-the-art 3D models provide all and more stimuli as employed in the presented study. The reductionist approach by Tsan et al. does however have a potential advantage, for example in high-throughput, but as to tissue quality limited, assays. This should be clearly stated as advantage and limitation of such models.

There are several important shortcomings in cardiomyocyte cultures on patterned substrate, e.g., 1) variable and often low cardiomyocyte seeding efficiency: please specify how many patterns per experiment were successfully seeded with 6-12 cardiomyocytes; also provide the inter-experiment variation;

2) how long (days, weeks, months?) can these cultures be maintained under baseline and stimulated conditions, with for example a positive inotropes or serum.

It must be clearly noted that traction forces are reported and that these forces do not represent the contractile forces measured in bona fide heart muscle by isometric force measurements. In fact, the statement in the Supplemental Notes, is fundamentally wrong: "Fibers were prescribed along the length of the M2DCT, which would lead to a contractility of up to 300 nN/ μm^2 (mN/mm²). This covers the range of typical myocardium contractility (~50 mN/mm²)." The contractile force in human myocardium is per cross-sectional and not per surface area. Unless the authors can provide an appropriate solution to this data conversion issue, any claims as to being in a similar range or representing normal myocardium must be deleted.

The deficiencies of standard, unorganized 2D cultures are well known. The authors introduce patterning to spatially confine 6-12 cardiomyocytes in a multicellular 2D culture assembly. This model was chosen to overcome the limitations in single cells grown on patterned substrates used by the authors (Ref 11) and others. A comparison of the multi-cell 2D assemblies to the advanced single cell models should be presented to clearly demonstrate an advance beyond the state-of-the-

art in 2D cultures.

The title must be rephrased: the method does not develop tissue; it does not provide compelling evidence for maturation beyond the state-of-the-art; it suggests an omission of stromal cells, which are never used or found advantageous in similar monolayer models. The title should reflect what the paper reports: a potentially improved 2D patterned cardiomyocyte culture

Minor:

All claims as to having tested dose-responses must be rephrased. Concentrations and not doses were tested. Dose testing can, per definition, only be done in vivo.

Supplement page 2: 30 μ M should read 30 μ m

Supplemental page 3: Table 1 is not included in the provided manuscript files

Reviewer #2:

Remarks to the Author:

Tsan et al have used 2-dimensional micro tissues to improve the organisation and maturity of pluripotent stem cell derived cardiomyocytes, a major goal of stem cell approaches to cardiac repair. The results are clearly presented and well illustrated and point to the importance of myofibrillar alignment and cell connectivity in driving highly reproducible differentiation and efficient contraction. The authors demonstrate the potential of this technique for screening and dissecting disease mechanisms. The following points should be addressed.

1. The authors show that multicellular connectivity is "associated with" robust myofibrillar development and function. Do alignment and contractions drive or accompany developmental improvements? Presumably there is feedforward activity here as enhanced differentiation further promotes performance.
2. Linked with this point, can the authors expand on the mechanisms by which their microculture parameters promote differentiation? For instance, can they test the importance of increased HOPX expression?
3. The protocol aims to minimize atrial cardiomyocyte differentiation. Do the authors observe heterogeneity in ventricular cardiomyocyte phenotypes? Is there any evidence for specialized conductive myocytes or other regional differences in ventricular cardiomyocyte programs observed in vivo?
4. The authors assess cardiac developmental endpoints to evaluate their protocol. How far do the authors consider their improvements have gone in promoting cardiomyocyte maturity compared to adult ventricular cardiomyocytes? Do the results provide any insight into the importance (or not) of exchange with other cell types (for example endocardium or cardiac fibroblasts) in vivo? Would addition of other cell types or signaling molecules further enhance maturation?
5. Please expand on the cell replication inhibition result on page 9.

Reviewer #3:

Remarks to the Author:

This manuscript describes an approach to develop a platform to provide organized and physiologically mature cardiomyocytes using micropatterning to capture small numbers of human pluripotent stem cell derived cardiomyocytes (hPSC-CMs). These hPSC-CMs then self-organise into small strips, called micron-scale two-dimensional cardiac tissues (M2DCTs), which develop properties such as alignment of myofibrils, increased expression of genes associated with maturing electrophysiological properties (eg. SCN5A, KCNJ2) and an action potential resembling mature

ventricular cardiomyocytes and punctate CX43 staining indicating the formation of intercalated discs. This platform was then used to examine the potential utility in drug testing using two drugs, the myosin ATPase inhibitor mavacamten and an L-type calcium channel inhibitor verapamil. For both drugs the M2DCTs showed an appropriate drug response and the authors' analysis suggests that this platform is more sensitive than non-patterned hPSC-CMs. To further drive maturation of M2DCTs a range of media compositions were trialled including dexamethasone, torin1, triiodo-L-thyronine (T3) or a media to increase oxidative phosphorylation (OxPhos). The OxPhos-T3 media was observed to induce further maturation by gene expression profiling of M2DCTs. Finally, M2DCTs' capacity in disease modelling was tested using a model in which MYBPC3 expression is perturbed due to a knockout of a promoter region. This analysis showed the hypercontractility observed in MYBPC3 related models of hypertrophic cardiomyopathy (increased contraction velocity and slower relaxation kinetics). Overall, the experiments are technically sound and well controlled, the manuscript is well written and figures are well presented. However, beyond the description of the M2DCTs the findings are broadly in line with those established in the literature. The M2DCT expands on well-established micro-patterning approaches for single cardiomyocyte analysis (previous work from this lab and e.g. Wang et al Nat. Med 2014) and comparable commercial alternatives exist for strips of cardiomyocytes (eg. Novoheart, 4DCell).

Other concerns.

RNAseq data permitted comparison with published data sets from 3D bioengineered heart tissue (e.g. Lam et al JAMA 2020, Mills et al Cell Stem Cell 2019, Ronaldson-Bouchard et al Nature 2018) to establish a comparison with the most mature human cardiomyocytes so far generated in vitro.

Fatty acid oxidation has recently been identified as a driver of maturation. Perhaps this could also be tested alongside the OxPhos media assayed in this manuscript?

To confirm the robustness of the M2DCTs a number of lines should be tested. Further, it would be informative to know how many of the 4,772 positions contained M2DCTs in each experiment.

We thank the reviewers for their careful critique of our work which has resulted in a much-improved manuscript. We have comprehensively addressed all the questions raised and have labeled responses consecutively in format of #1, #2, etc. to cross-reference similar responses across reviewers. Highlights of the revision include the following:

- Improved clarity of comparisons and relative development (maturation) statements throughout with improved discussion on advantages/disadvantages of 2DMBs (#1, #2, #8)
- New comparison with single micropatterned hPSC-CM technique, including new analyses and videos to demonstrate the high yields of 2DMBs using our methods (#4, #17, #20)
- New supplemental videos 4-5 showing live-cell imaging of myofibrillar alignment and intercalated disks with a new reporter desmoplakin-GFP (#4, #17)
- New comparison with 3D cardiac tissue RNA-seq data (#2, #18)
- Re-organized and additional RNA-seq analyses, including 1) adding 3D cardiac tissue comparison, 2) new analyses of cell proliferation pathways and upstream regulators of transition from proliferative potential state toward terminal differentiation state, 3) splitting figure 5 into new figures 5-6 with separate result sections to improve clarity on which developmental endpoints are most driven in 2DMBs vs. metabolic/hormonal cues (#2, #12, #13)
- New mechanistic experiment linking contractile force causally to myofibrillar growth/development in new Figure 7, also demonstrating ventricular lineage marker MLC-2v (#12, #14)
- New mechanistic experiment linking calcium load causally to sarcoplasmic reticula development using SERCA2-GFP reporter line (# 12)
- Demonstration of generalizability of the methods in 5 new iPSC lines, including the SERCA2-GFP line, desmoplakin-GFP line, and 3 additional randomly selected control hPSC lines, bringing the total number of separate hPSC lines in the study to 9 (#20)

Reviewer 1 (Remarks to the Author):

Tsan and colleagues present an advanced monolayer cardiomyocyte culture method, comprising ensembles of 6-12 cells on micropatterns. Patterns on soft (8 kPa) fibronectin-coated PDSM substrates with 7:1 (308 μm x 45 μm) dimensions are used instead of the often employed smaller (e.g., 70 μm x 10 μm) single cell sized patterns. The increase in pattern size overcomes the well-known limitations associated with single cardiomyocyte growth. Ensemble cardiomyocyte cultures on large patterns were compared to “standard” monolayer cultures without spatial constraints. The study applies a broad range of methods, including numerical simulations, electrophysiology, RNA-seq, mass spectrometry. While an advance over “standard” monolayer cultures with a well-known poor degree of maturation is plausible and expected, there is no data in support of an advance over single cells on patterns or 3D culture models. The authors should be more specific as to the real advantage of their model, which is very likely not maturity, but could be applicability in high throughput assays with an increase in robustness compared to single cardiomyocytes on patterns.

#1 We thank the reviewer for his/her careful critique of our work. We agree that we showed that our technique is a major advance over standard monolayer culture, while it also maintains advantages of 2D culture for imaging and fluorescence read-outs. The major goal was to improve physiologic and measurement reproducibility of contractile function in an iPSC-CM system that would be accessible for most labs. Reproducible assessment of contractile function remains a major bottleneck, and this limitation has not been resolved by either single cell methods (as we have recently shown, PMID

33577793), or by 3D tissue systems (e.g. 33053362). To accomplish this primary goal, we began with concepts developed from 3D tissues, and reverse-engineered the key elements that we hypothesized were, at least in part, responsible for the greater maturation observed in 3D tissues – in particular, the physiologic elasticity and uniaxial contractile direction present in these systems. The 2DMB platform we developed was not intended to result in superior maturation than 3D tissues, and we have been careful to not make any such claim. On the contrary, the platform will enable labs without sophisticated 3D tissue set-ups to efficiently and accurately measure contractile function in a tractable system that has the potential to increase rigor in the field (all of our algorithms will be made freely available on GitHub). We agree that a direct comparison with single cell micropatterning is important and, in the revision, have directly compared 2DMBs to single micropatterned hPSC-CMs, demonstrating that the 2DMB method results in markedly improved throughput with greater reproducibility in myofibrillar development (see #4 and #17 below). We also agree that a direct comparison with 3D tissues was needed to demonstrate where on the developmental spectrum 2DMBs are positioned and have added a transcriptomic comparison with advanced 3D tissues (see #2 below).

Thank you for the suggestion to add clarity about the major advantages and disadvantages of 2DMBs. We have added clarity throughout the manuscript on these points. We agree that a major advantage of our method is the simplified and reductionist approach and have highlighted this in the new title, abstract, and elsewhere. We have also added a limitations section that describes some of these issues with model selection for different types of studies and where 2DMBs are an advantage. A major finding of our work is that 2DMBs do, in fact, progress maturation far beyond standard 2D monolayer hPSC-CMs, and these findings of high relevance for disease modeling and other applications where immaturity may limit applicability for human disease. We have refined much of the results on these points, including additions of comparisons to single micropatterned hPSC-CMs and 3D cardiac tissues (details in #2, #4). Specifically, we found that 2DMBs exhibit: 1) increased myofibrillar density compared to single cells (new Figure 2G), 2) increased contractile function compared to single cells, 3) physiologic sarcomere length of 2.2 μm compared to single cells (Figure 1), 4) improved myofibrillar alignment (Figure 1), 5) marked up-regulation of critical ion channel genes *SCN5A*, *KCNJ2*, *GJA1* (and others) with corollary reduction in *HCN4* compared to standard monolayers – findings that correlated with a relatively mature ventricular AP profile, -80 mV resting membrane potential, and calcium handling (Figure 3), 6) gene expression pattern profile consistent with maturation across multiple parameters (Figure 5), and shift to adult myofibrillar gene isoforms (Figure 5-6, suppl figure 9). All of these findings are considered important indicators of maturation along the spectrum of cardiac development (PMID 31388700). Notably, in the transcriptomic analysis, we also compared to other recently published techniques that have achieved varying levels of maturation in 2D monolayers, including metabolic manipulation through media formulation, thyroid hormone, dexamethasone, and torin1 (as reviewed, PMID 31388700) – we validate consistent findings across these other techniques, but show far more progression along the maturation trajectory in 2DMBs (Figure 5A-C). Thus, we provide compelling evidence that the physiologic mechanical and environment of 2DMBs is sufficient to drive maturation well beyond that attained in standard 2D culture, with or without additional maturation cues, or single micropatterned iPSC-CMs. Please see further below regarding the new comparison to 3D tissues (#2). We also performed additional experiments demonstrating that contractile force in 2DMBs is necessary to achieve the attained state of development (#12 below).

We have revised parts of the introduction and discussion to add clarity on the advantages of the 2DMB technique – specifically that it improves throughput and reproducibility in an imminently tractable 2D culture model through both uniformity of the contractile environment and further progressed cardiomyocyte development, to the extent possible, in 2D.

Main critiques:

All claims as to an improvement over 3D models must be deleted, unless the authors include a state-of-the-art 3D model for direct comparison in their study. The study is essentially on 2D with a nice demonstration that ensembles of cells rather than individual cells may have advantages as to applications in drug testing and disease modelling. It is highly unlikely and not supported by any data, that maturity in 2D cultures, irrespective of their format, would reach closely the degree of maturation that can be achieved in state-of-the-art 3D models. For clarity, all wording suggesting a tissue model must be deleted.

#2 We agree with the Reviewer that 2DMBs would not be expected to improve maturation beyond 3D tissues and have not in the original submission, nor in the revision, made any claim as such. The previously proven maturation in 3D tissues was, in fact, the motivation for the 2DMB platform, which simplifies elements of the contractile environment of 3D microtissues (and in vivo cardiac muscle) down to a planar, elastic substrate. Our original submission included a statement in the introduction to provide this motivation and highlighted the maturation evident in 3D tissues, i.e. “[3D] cardiac tissues suspended between elastomer posts allow physiologic uniaxial contractions and induce maturation”. To emphasize the proven maturation shown with 3D tissues, we have added an additional phrase in the beginning of the next sentence “While clearly increasing maturation” before providing the motivation for a 2D system that captures some elements of the 3D tissue environment (uniaxial contractile direction and geometric alignment cues) to create the simplified/reductionist 2DMB platform.

To place our new platform in the context of 3D tissues, we have added a comparison of RNA-seq data from 2DMBs to 3D organoid tissues (Mills, et al. *PNAS*, PMID 28916735). We chose a dataset (as suggested by Reviewer 3) that included RNA-seq data processed by a highly similar method as our prior data. The 3D cardiac tissue dataset also included an adult left ventricular sample, enabling comparison across the full developmental spectrum. See Results p11 and Figure 5. In summary, we show that a physiologic mechanical environment in 2D is in fact sufficient to achieve several cardiac developmental milestones, as described above (#1). Interestingly, some genes considered maturation markers, (e.g. *SCN5A*, *KCNJ2*) were similarly expressed in 2DMBs as in 3D cardiac organoids. As expected, 3D tissues, possibly by nature of fibroblast signaling, reach a higher myofilament maturation potential, as evidenced by further up-regulation of adult myofilament isoforms (e.g. *MYL2*, *TNNI3*). Interestingly, key genes regulating the transition from the cardiac progenitor state toward terminal cardiomyocyte differentiation (e.g. *HOPX*, *WNT2*, *TBX2*), were similarly expressed in 2DMBs, cardiac organoids, and adult human heart tissue (Results p11, Figure 5E). This additional analysis and commentary significantly improved the manuscript, by clearly describing where on the developmental spectrum 2DMBs reside, and more clearly highlighting the applications for which 2DMBs may hold advantages as a more reductionist system.

We agree with the Reviewer’s description of our platform as “ensembles of micropatterned cells”. Although an ensemble of similar cells and associated ECM (in this case deposited through micropatterning plus whatever is extruded by the cells themselves) meets the minimal definition of “tissue”, we have changed the terminology throughout to refer to these assemblies as 2D cardiac muscle bundles (2DMBs). To clearly demonstrate the point that 2DMBs are functionally discrete and independent, we have added Supplemental Video 2, in which 2DMBs can be seen as independently and autonomously contracting, each mechanically isolated from adjacent 2DMBs. Needless to say, we did not intend to indicate that the platform we have developed would supplant 3D tissues, which are

obviously immensely useful for the understanding of cardiac development and physiology in a complex, mixed cell type environment. Rather, 2DMBs are useful as an adjunctive technique to study contractile behavior and development in a highly controlled and reductionist environment as part of the armamentarium available for researchers using the hPSC-CM model system.

The claim that the model overcomes the use of stromal cells is not used within the right context. In monolayer cultures, such as presented by the authors, non-myocytes have never been a requirement. In fact, in such models, non-myocyte number is always kept low to avoid overgrowth. Even in 3D models, stromal cells are not an absolute requirement for individual cardiomyocyte seeding and function. In 3D models, stromal cells are required to organize a complex cardiomyocyte maturation supporting biological [including, but not limited to fibronectin]/biophysical [8-12 kPa] ECM environment. As such, state-of-the-art 3D models provide all and more stimuli as employed in the presented study. The reductionist approach by Tsan et al. does however have a potential advantage, for example in high-throughput, but as to tissue quality limited, assays. This should be clearly stated as advantage and limitation of such models.

#3 Thank you – we agree and did not intend a claim that the model “overcomes” the use of stromal cells. To prevent this unintended emphasis, we have removed “without stromal cells” from the title. We further agree that the primary advantage of the 2DMB system is as a reductionist method that rigorously controls the contractile environment and improves throughput. In 2DMBs, a lack of requirement for stromal cells removes one source of variability that, although they may enhance maturation, also may influence batch-batch and line-line variability and confound disease modeling studies (e.g. PMID 33053362), particularly since successful 3D tissue generation has been reported to be highly dependent upon the proportion of contributing fibroblasts (e.g. PMIDs 24255110, 29618819). The high level of geometric and cell composition control of 2DMBs, as well as the automated and blinded analytic tools that we developed for this study, provide a tractable solution to increase reproducibility in contractile measurements for disease modeling at the developmental window attained by 2DMBs. We have revised the title, clarified the context in the introduction better, and added relevant statements in the discussion to further clarify these important points, including a new limitations paragraph (p16).

There are several important shortcomings in cardiomyocyte cultures on patterned substrate, e.g., 1) variable and often low cardiomyocyte seeding efficiency: please specify how many patterns per experiment were successfully seeded with 6-12 cardiomyocytes; also provide the inter-experiment variation;

#4 Thank you – this is an important point that needed further clarity in the manuscript. We agree with the Reviewer that single cell seeding on micropatterns has very low efficiency. For example, Alexandre Ribeiro, the lead author on the most comprehensive single cell report to date, PMID 26417073, reports that >75% of single iPSC-CMs are “defective” after seeding micropatterned substrates, see <https://www.fda.gov/media/126127/download>, and prior reports using this technique have generally had low numbers of replicates and very large variance in measurements (e.g. PMID 26417073). A major motivation for the present study was that we also found a disconcerting level of variability in the individual cell contractile phenotypes due, in part, to stalled and variable myofibrillar development across single micropatterned iPSC-CMs, which we recently published (PMID 33577793).

In the revision, we have directly compared to single micropatterned hPSC-CMs. We show that the 2DMB method is far superior to single micropatterned iPSC-CMs in terms of 1) attachment yield, 2) proportions of analyzable replicates, 3) myofibrillar density in resultant 2DMBs, 4) contractile performance, 5)

reduced variance (Results p7, Figure 2G, Supplemental Video 2 vs. 5). Development of this method for the more strongly contracting 2DMBs required several optimizations that also ultimately resulted in a far faster and more efficient workflow than possible with the single cell traction force method (see also #17 below, new Supplemental Figure 4). We are well-qualified to make these comparisons, having spent years optimizing and utilizing single cell iPSC-CM techniques (PMIDs 31877118, 33577793), including the development of an automated and blinded image processing algorithm that we employed to quantify and compare myofibrillar density across both methods to avoid potential for bias (PMID 33577793). This analysis demonstrated a significant increase in myofibrillar density in 2DMBs (Figure 2G). Moreover, whereas single micropatterned iPSC-CMs exhibit extensive variability in myofibrillar density, 2DMBs demonstrated a nearly 50% reduction in the coefficient of variation in this measurement, indicating that more robust myofibrillar development paralleled reduced replicate variability (Figure 2G). Additionally, we compared uniaxial force generation of 2DMBs vs. single micropatterned iPSC-CMs (normalized to width) and found an 80% increase in force generation for 2DMBs (Results p7). Finally, we showed that the 2DMB method is generalizable across diverse hPSC lines (Suppl. Figure 5, #20 below). As an additional practical advantage, the high yield of micropatterning with the 2DMB method readily enables biologic assays from pooled samples, such as the RNA-seq comparison in the initial manuscript, that are not generally feasible in single iPSC-CMs without expensive and inefficient single-cell quantification approaches.

Another application of the method is the ability for quantification and live-monitoring of cell junctions, which we add demonstrated previously in Supplemental Figure 2. In the revision, we have also included videos showing this point in Supplemental Videos 4-5. Quantification of cell junctions clearly is not possible with single micropatterned hPSC-CMs.

2) how long (days, weeks, months?) can these cultures be maintained under baseline and stimulated conditions, with for example a positive inotropes or serum.

#5 For the present study, we analyzed all 2DMBs at 8-10 days after plating (which was 32 days from initiation of differentiation), at which time 2DMBs exhibit robust myofibrillar alignment, intercalated disk and gap junction assembly, calcium handling machinery (see also new Figure 6D-F for additional SERCA2 data), and contractile function. Our routine culture conditions for 2DMBs included a more physiologic calcium level of 1.45 mM, which has a clear inotropic effect relative to the typical 0.4 mM calcium of the RPMI base media used by many labs (see Figure 4 for calcium concentration response). We have observed that in high calcium, beyond 2 weeks, that some 2DMBs do augment contractile force enough to break their focal adhesion connections to the PDMS substrate. We have not tested serum due to its pleiotropic effects. The main applications we foresee of 2DMBs will be as a simplified model testbed for disease modeling and acute/subacute hypothesis testing in the setting of a controlled biomechanical environment, and the 32-day time point works well for these applications. We have added discussion of this important point in the new limitations paragraph (p16).

It must be clearly noted that traction forces are reported and that these forces do not represent the contractile forces measured in bona fide heart muscle by isometric force measurements. In fact, the statement in the Supplemental Notes, is fundamentally wrong: "Fibers were prescribed along the length of the 2DMB, which would lead to a contractility of up to 300 nN/ μm^2 (mN/mm²). This covers the range of typical myocardium contractility (~50 mN/mm²)." The contractile force in human myocardium is per cross-sectional and not per surface area. Unless the authors can provide an appropriate solution to this data conversion issue, any claims as to being in a similar range or representing normal myocardium must be deleted.

#6 We thank the reviewer for the comment relating to the modeling method description. We have now added further clarifications to the Supplemental Notes to explain that the main purpose of the modelling study was to capture the underlying PDMS response to forces exerted by an 2DMB, which was modelled to cover the range of deformations we might expect in vivo. As the reviewer states, the stress in these tissues is a force through cross-sectional area (to the fibers, assuming no dispersion). The reported value of up to 300 nN/ μm^2 was applied to the modelled tissue strip internally, representing an active contraction within the strip itself (force per unit cross-sectional area), to establish the modeling parameters, not as an output of force estimation. This parameter was used to investigate the net force on the PDMS itself, which was reported as the total force integrated over the surface area of contact between the muscle bundle and the PDMS (traction force). This represents the cumulative applied force. Note that the total force reported must be equivalent to the integral of the traction (Cauchy stress dotted with the normal vector of the cross-section) across the cross-section in the middle of the tissue (midway between the two blue boxes in Figure 2B). In the main text, we reported the total traction force. As the Reviewer points out, it is difficult to directly compare force between a 2D system and 3D tissues, and we did not previously, or in the revision, attempt to do this. As a separate, but related point, we have added a comparison of traction force generated by 2DMBs vs. single micropatterned iPSC-CMs as described above.

The deficiencies of standard, unorganized 2D cultures are well known. The authors introduce patterning to spatially confine 6-12 cardiomyocytes in a multicellular 2D culture assembly. This model was chosen to overcome the limitations in single cells grown on patterned substrates used by the authors (Ref 11) and others. A comparison of the multi-cell 2D assemblies to the advanced single cell models should be presented to clearly demonstrate an advance beyond the state-of-the-art in 2D cultures.

#7 Please see #5 above

The title must be rephrased: the method does not develop tissue; it does not provide compelling evidence for maturation beyond the state-of-the-art; it suggests an omission of stromal cells, which are never used or found advantageous in similar monolayer models. The title should reflect what the paper reports: a potentially improved 2D patterned cardiomyocyte culture

#8 We demonstrate that physiologic biomechanics in 2DMBs are sufficient to strongly influence cardiac development across a range of endpoints, including electrophysiology, contractile performance, and transcriptional regulation (please see #1-2 above). Thank you for the suggested emphasis on the reductionist nature of this platform, which we have added to the revised title. More than just reporting on a new model with careful multimodality validation, we have also investigated early developmental signals influenced by the contractile environment and intersecting hormonal/metabolic cues, for which we have improved the clarity of the results section (p10-12) and split Figure 5 into improved new Figures 5-6. Additionally, we have added mechanistic experiments that shows a causal relationship between contractile force and myofilament growth (Figure 7), as well as between calcium load and sarcoplasmic reticula development (Figure 7), both of which were enabled by the 2DMB platform (see #12 below). These findings have important relevance transcendent across hPSC-CM model systems.

We have, in tandem, created automated image-based quantification tools that we will make freely available on GitHub upon publication, since we do not intend to commercialize any aspect of this method. The materials and methods developed here could be readily implemented in any laboratory with basic hPSC-CM expertise (except for fabrication of the initial silicon wafer used to generate micropatterning stamps, but these can be obtained commercially and can be re-used nearly indefinitely – we will also make the CAD file for fabrication by commercial vendors freely available). In summary, the

2DMB technique markedly increases throughput for contractile studies, and we anticipate that the approach will be useful for a large and growing number of labs interested in hPSC-CM contractile studies and who 1) have struggled with single cell approaches, 2) have experienced too much heterogeneity with non-physiologic monolayer approaches, 3) would benefit from a time- and cost-effective approach for contractile quantification for disease modeling, 4) do not have bandwidth or expertise for 3D tissue approaches, or 5) desire a reductionist platform for mechanistic studies in a stable contractile environment. We have improved the clarity of the text to highlight these points and have made the advantages of this system clearer.

Minor:

All claims as to having tested dose-responses must be rephrased. Concentrations and not doses were tested. Dose testing can, per definition, only be done in vivo.

#9 Thank you – this correction has been made.

Supplement page 2: 30 μ M should read 30 μ m

#10 Thank you- this correction has been made.

Supplemental page 3: Table 1 is not included in the provided manuscript files

#11 Thank you – we have included Table 1.

Reviewer #2 (Remarks to the Author):

Tsan et al have used 2-dimensional micro tissues to improve the organisation and maturity of pluripotent stem cell derived cardiomyocytes, a major goal of stem cell approaches to cardiac repair. The results are clearly presented and well illustrated and point to the importance of myofibrillar alignment and cell connectivity in driving highly reproducible differentiation and efficient contraction. The authors demonstrate the potential of this technique for screening and dissecting disease mechanisms. The following points should be addressed.

1. The authors show that multicellular connectivity is "associated with" robust myofibrillar development and function. Do alignment and contractions drive or accompany developmental improvements? Presumably there is feedforward activity here as enhanced differentiation further promotes performance.

#12 Thank you for raising this important point. We also speculated a feedforward mechanism whereby uniaxial contractile behavior in organized tissues enables continued myofibrillar growth and development, which would be rational from a teleologic perspective given that in vivo cardiac development occurs during a gradually increasing pulsatile workload. To test whether contractile function is itself required for the myofilament development observed in 2DMBs, we subjected 2DMBs to the cardiac myosin-ATPase inhibitor mavacamten for 8 days prior to assay. Since mavacamten is a specific cardiac contractile inhibitor without apparent off-target effects (with recent clinical trials published and currently under FDA review), this experiment enabled us to isolate the impact of contractile force on myofilament development. In this new experiment, we demonstrate that reduction in contractility markedly reduces the key myofilament maturation regulator *HOPX*, as well as key myofilament gene expression (*MYH7*, *MYL2*), see Results p12-13, Figure 7. We further demonstrate a marked reduction in MLC-2v (encoded by *MYL2*) abundance in myofilaments with reduction in

contractile force (Figure 7B-C). Together, these new data demonstrate that the greater uniaxial contractile force generated in 2DMBs is critical for myofilament maturation and growth.

Linked with the importance of contractility, we also tested the importance of calcium concentration during development in 2DMBs. In the original submission, we had used an optimized calcium concentration that improved contractile function (Figure 3) but did not investigate how the calcium concentration might impact myofilament and calcium handling development. Notably, many labs using hPSC-CMs currently use an RPMI-based media, which has a low calcium concentration of 0.4 mM. In the revision, we show that increasing the calcium concentration to near-physiologic levels (1.45 mM), is critical for development of sarcoplasmic reticula in 2DMBs, which we demonstrate by using a SERCA2-GFP reporter iPSC line. In this experiment, we again used mavacamten to create a similar level of contractile blunting in the high calcium condition, demonstrating that calcium itself is critical for sarcoplasmic reticula development, distinct from the contractile force, while calcium-induced increased contractile force also contributes to myofilament growth (Figure 7D-F).

2. Linked with this point, can the authors expand on the mechanisms by which their microculture parameters promote differentiation? For instance, can they test the importance of increased HOPX expression?

#13 Thank for raising this point. We demonstrate in the new experiment above that contractile force is causally linked to myofilament growth and maturation in 2DMBs. *HOPX* is a key regulator of terminal muscle cell differentiation, and *HOPX* was previously shown to be arrested in standard iPSC-CM monolayer cultures and causally linked to myofilament growth (PMID 30290179). In our prior analysis that *HOPX* was increased by 12-fold in 2DMBs, and this increase was blunted in the presence of mavacamten (new Figure 7). We further investigated downstream genes of *HOPX* in the RNA-seq data, and we found that genes directly repressed by *HOPX* were reduced (e.g. *WNT2*, *AXIN2*). These genes are critical for maintain proliferative potential in cardiac progenitors and immature cardiomyocytes; hence their reduction indicates a progression toward terminal differentiation. The transition toward a terminally differentiated state is also supported by a reduction in *TBX2*. Notably, the pattern of *HOPX* increase and *WNT2-AXIN2-TBX2* decrease was similar in 2DMBs as in 3D cardiac organoids and adult human heart in a new analysis comparing data from Mills et al. (PMID 28916735), as shown in new Figure 5E. Further corroborating a shift toward terminal differentiation, we show through additional pathway analysis of the RNA-seq data that DNA replication and cell proliferation genes are markedly down-regulated in 2DMBs under the optimized media conditions we used (Results p11, Figure 5C, Supplemental Figure 8).

3. The protocol aims to minimize atrial cardiomyocyte differentiation. Do the authors observe heterogeneity in ventricular cardiomyocyte phenotypes? Is there any evidence for specialized conductive myocytes or other regional differences in ventricular cardiomyocyte programs observed in vivo?

#14 This is a great point – we attempted to create a controlled lineage direction based on the published protocols to date. Since retinoic acid signaling has very clearly been linked to atrial lineage specification in many studies, we used CDM3 (which is retinol-free) for cardiac induction at day 3 of differentiation, additionally supplemented with retinol inhibitor to block any residual endogenous signaling. Patch-clamping of 2DMBs showed action potentials that reproducibly fit a ventricular profile (Figure 3). In new Figure 7, we included immunofluorescence quantification of the ventricular isoform of myosin light chain (MLC-2v), since this isoform is associated not only with ventricular lineage specification, but also ventricular maturation. As seen in the figure, MLC-2v was present in myofilaments of nearly all hPSC-

CMs in 2DMBs, indicating ventricular lineage. Not surprisingly, the expression level across individual cells was variable, likely suggesting that the extent of maturation in individual cells is variable. Additionally, there could be other ventricular lineage subtypes present. A detailed per-cell analysis of subtype lineages is beyond the scope of the current study, but we agree that this is a key question, as uniformity of cell types would heighten relevance for both disease modeling and therapeutic applications. The 2DMB platform will be an excellent tool to further dissect lineage subtypes, since it is readily amenable to the use of reporter lines (as we showed with connexin-43 and SERCA2), minimizes the confounding influence of contractile force (which we show to be critical in new Figure 7), and enables per-cell quantification with standard 2D immunofluorescence.

4. The authors assess cardiac developmental endpoints to evaluate their protocol. How far do the authors consider their improvements have gone in promoting cardiomyocyte maturity compared to adult ventricular cardiomyocytes? Do the results provide any insight into the importance (or not) of exchange with other cell types (for example endocardium or cardiac fibroblasts) in vivo? Would addition of other cell types or signaling molecules further enhance maturation?

#15 Thank you – this is an important point. Our goal in this project was not necessarily to generate the most mature hPSC-CMs possible, but rather to create a highly controlled microenvironment that would progress hPSC-CM maturation into a relatively stable developmental window to improve fidelity of contractile studies, as well as the type of reductionist study that we have added with the new mavacamten and calcium experiments (Figure 7). Please see #2 above, where we describe the addition of an RNA-seq data comparison of 2DMBs, 3D cardiac tissues, and adult human heart. Yes, we agree that it is highly likely that other cell types (particularly fibroblasts) provide developmental cues. The fibroblast secreted extracellular matrix is essential for proper terminal differentiation of chamber specific cardiomyocyte lineage subtypes, and the specific ECM ligand-integrin interactions that govern these processes are only beginning to be understood. As Reviewer 1 points out, there are many and various inputs affecting the biology of 3D tissues, and parsing these various inputs to dissect developmental signals is challenging. 2DMBs can be used as a complimentary system to quantify effects of individual inputs, with our focus in the present study being on the contractile environment itself. Future work could, for example, test candidate pathways linking fibroblast-mediated integrin signaling gleaned from studies in 3D tissues (or whole hearts) to test causal mechanisms, adding to the armamentarium available for iPSC-CM mechanistic studies.

5. Please expand on the cell replication inhibition result on page 9.

#16 Thank you – cell proliferation is a key marker of transition out of the cardiac progenitor state, as shown both in the paper previously referenced (in which torin was used to block cell replication, PMID 31707831) and in a recent paper showing that low-dose activation of wnt signaling can result in “massive” expansion in a maintained cardiac progenitor state (PMID 32619518). *HOPX*, which was previously shown to be stalled in standard monolayers, may be a major nodal regulator of transition from a proliferative state toward terminally differentiated muscle cells (PMID 30290179). In the revision, we have expanded our analysis of the transcriptional regulation of cell proliferation using our prior RNA-seq data set, including a new pathway analysis focused on DNA replication and cell proliferation (Figure 5C, Supplemental Figure 8, Results p11). Interestingly, we show that in 2D monolayers, shifting toward oxidative phosphorylation and/or adding T3 is sufficient to drive some regulatory shift away from the proliferative state, similar to inhibition by torin1 (which was more modest in our study). The physiologic biomechanical environment of 2DMBs demonstrated an even stronger shift, with a marked reduction of cell replication genes and a marked increase in negative cell proliferative regulators (Figure 5C). These data strongly support a shift from the cardiac progenitor state toward a terminally differentiated non-proliferative state (Figure 5E, see also #13 above).

Reviewer #3 (Remarks to the Author):

This manuscript describes an approach to develop a platform to provide organized and physiologically mature cardiomyocytes using micropatterning to capture small numbers of human pluripotent stem cell derived cardiomyocytes (hPSC-CMs). These hPSC-CMs then self-organise into small strips, called micron-scale two-dimensional cardiac tissues (2DMBs), which develop properties such as alignment of myofibrils, increased expression of genes associated with maturing electrophysiological properties (eg. SCN5A, KCNJ2) and an action potential resembling mature ventricular cardiomyocytes and punctate CX43 staining indicating the formation of intercalated discs. This platform was then used to examine the potential utility in drug testing using two drugs, the myosin ATPase inhibitor mavacamten and an L-type calcium channel inhibitor verapamil. For both drugs the 2DMBs showed an appropriate drug response and the authors analysis suggest that this platform is more sensitive than non-patterned

hPSC-CMs. To further drive maturation of 2DMBs a range of media compositions were trialled including dexamethasone, torin1, triiodo-L-thyronine (T3) or a media to increase oxidative phosphorylation (OxPhos). The OxPhos-T3 media was observed to induce further maturation by gene expression profiling of 2DMBs. Finally, 2DMBs capacity in disease modelling was tested using a model in which MYBPC3 expression is perturbed due to a knockout of a promoter region. This analysis showed the hypercontractility observed in MYBPC3 related models of hypertrophic cardiomyopathy (increased contraction velocity and slower relaxation kinetics). Overall, the experiments are technically sound and well controlled, the manuscript is well written and figures are well presented. However, beyond the description of the 2DMBs the findings are broadly in line with those established in the literature. The 2DMB expands on well-established micro-patterning approaches for single cardiomyocyte analysis

(previous work from this lab and e.g. Wang et al Nat. Med 2014) and comparable commercial alternatives exist for strips of cardiomyocytes (eg. Novoheart, 4DCell).

#17 Thank you for reviewing our work. Despite micropatterning techniques being applied to contractile studies in cardiomyocytes several years ago (e.g. in NRVMs, PMID 23159216), there are relatively sparse publications using this technique for contractile studies in hPSC-CMs in the literature to date, and those that exist have relied on low replicate numbers. This is likely due to 1) the low seeding efficiency of single micropatterned iPSC-CMs (see #4 above), 2) the low throughput of these single cell techniques, which have relied on traction force microscopy to reliably track motion of single cells, and 3) the heterogeneity and stalled myofibrillar development in single iPSC-CMs as we recently showed (PMID 33577793). In the revision, we have added a direct comparison to the single cell technique, clearly showing the superiority of seeding efficiency and reproducibility of 2DMBs (see #4). Creation and optimization of the technique also required switching the elastic substrate from polyacrylamide gels to physiologic stiffness PDMS formulation (8 kPa), a relatively simple but important discovery, since 2DMBs contracted with too much force to maintain adhesion on functionalized polyacrylamide gels (see new Supplemental Videos 1-2) – note that this also required optimization of a different micropattern transfer technique that proved to be simpler and faster than prior the methods that we (PMID 33577793, 31877118) and others (e.g. PMID 26417073) have used on polyacrylamide gels, creating a much faster and simpler overall process. Additionally, 2DMBs were too large for practical implementation of traction force microscopy quantification (which is computationally intensive / time consuming with ~45 minutes processing per single cell per beat on a high-end workstation), so we also developed a very

efficient image processing algorithm to calculate longitudinal fractional shortening in 2DMBs and convert these measurements to traction forces (<20 seconds computational time per 2DMB for 4-5 beats), enabling rapid and automated batch quantification. We also integrate these methods with our recently described myofibrillar analysis method (PMID 33577793), such that batches of 2DMBs can be quantified for both contractile function and myofibrillar structure to verify similarity across muscle bundles and batches, enabling a high level of quality control that will increase the precision of future contractile studies. Another application of 2DMBs is the ability for quantification and live monitoring of cell junctions in a controlled contractile environment, which we had demonstrated previously in Supplemental Figure 2. In the revision, we have also included videos showing this point in Supplemental Videos 4-5 using live cell visualization in a desmoplakin-GFP reporter line. In summary, we have developed a comprehensive technique that efficiently and cost-effectively generates large yields of high quality 2D contractile cardiac muscle bundles on elastic substrates that can be analyzed with fully automated and fast image processing tools. We have summarized the improvements in workflow for the 2DMB method in new Supplemental Figure 4.

Only a single other 2D technique, to our knowledge, replicates physiologic and uniaxial contractile shortening in anisotropic 2D cardiac tissue, which is the muscle “flap” technique developed in Kit Parker’s lab – this technique has been useful to understand physiologic abnormalities in Barth syndrome and in CPVT (PMIDs 2481325, 31311300), but is limited in throughput and requires dedicated equipment (laser cutter, etc.) for fabrication. The commercially available micropatterning services cited are not relevant for the patterning of soft (8 kPa) PDMS used here, and certainly none of these current commercial resources, to our knowledge, offer the comprehensive contractile platform we have carefully developed and validated here.

Beyond method development, comparison to other 2D maturation approaches (Figure 5), and application for pharmacologic and disease modeling, we have additionally applied the 2DMB platform to understand intersecting cues during early cardiomyocyte development and to perform a mechanistic analysis of the role of contractile function and calcium concentration in influencing cardiac development in response to Reviewer 2’s comment (see #12 above). Taken together, these revisions substantially raise the impact of our work and greatly appreciate the reviewers’ critiques.

Other concerns.

RNAseq data permitted comparison with published data sets from 3D bioengineered heart tissue (e.g. Lam et al JAHA 2020, Mills et al Cell Stem Cell 2019, Ronaldson-Bouchard et al Nature 2018) to establish a comparison with the most mature human cardiomyocytes so far generated in vitro.

#18 Thank you for this excellent suggestion. Of the 3 studies, the one by Mills et al (PMID 28916735) provides the best comparison data set, since a highly similar RNA-seq analysis pipeline was used and the 3D tissues were well-validated (Ronaldson-Bouchard et al. used microarrays so the gene expression data is not directly comparable). As discussed above (#2), we found that several genes considered markers of maturation were similarly expressed between 2DMBs and cardiac organoids (e.g. *SCN5A*, *KCNJ2*), while the myofilament adult isoforms *MYL2* and *TNNI3* were more highly expressed in cardiac organoids (though still much lower than in adult heart). Taken together, these data show that some aspects of transcriptional regulation in 2DMBs are similarly progressed as in 3D tissues/organoids, while myofilament isoform maturation is incrementally further progressed in 3D tissues/organoids.

Fatty acid oxidation has recently been identified as a driver of maturation. Perhaps this could also be tested alongside the OxPhos media assayed in this manuscript?

#19 Thank you – we agree that there are intriguing recent reports regarding fatty acid oxidation. We

tested a media modified from a report by Correia et al. (PMID 28819274). Interestingly, that study reported a greater level of maturation with fatty acid supplementation, but most of the developmental improvements were already present in a comparison group in which lactate preconditioning (without fatty acids) was used to mimic fetal cardiac metabolism (which is lactate-dependent). In our pilot testing, we included various versions for Correia et al.'s media, including with or without additional fatty acids, and did not see an apparent difference in contractile function. Since we showed a clear maturation effect with switch from sTnI to cTnI with the version we referred to as "OxPhos" that did not include fatty acids beyond a small amount included in the B27 supplement (Figure 6A-B), we proceeded with that formulation. We did not exhaustively test all fatty acid formulations published in the literature – the specific formulation, vendor, and preparation may all be important since fatty acids are prone to oxidation, which can also present the possibility of cell toxicity. We agree that further study of fatty acids in light of more recent publications would be an interesting future direction.

To confirm the robustness of the 2DMBs a number of lines should be tested. Further, it would be informative to know how many of the 4,772 positions contained 2DMBs in each experiment.

#20 Thank you – we agree that robustness across several lines would be a strong indication of broad applicability and verification of this was much needed in the manuscript. In the revision, we have added an analysis of 2DMBs generated from 3 randomly selected control hPSC lines from healthy individuals that were available through an MTA at our university and have shown reproducibility of the method across all of these lines (Supplemental Figure 5). Additionally, in the revision we have also included analyses in an additional GFP reporter line (SERCA2-GFP) from the Allen Institute (Figure 7) and visualization of intercalated disks with another new GFP reporter line (desmoplakin-GFP, Supplemental Videos 4-5). Together, this brings the total number of separate iPSC lines in the study to 9 (DF19 line, DF19 MYBPC3 gene-edited line, WTC line, connexin-43-GFP line, SERCA2-GFP line, desmoplakin-GFP line, 3 randomly selected control lines). We generally obtain a very high yield of 2DMBs per substrate with this technique – this high yield not only makes 2DMBs time- and cost-effective, but also amenable to biochemical assays from pooled samples. We have added the quantification of yield of 2DMBs in the comparison to the single cell technique as described in #4 (see Results p7, Supplemental Video 2).

Reviewers' Comments:

Reviewer #1:

Remarks to the Author:

The authors have addressed my critiques. The study adds important insight and a novel patterning strategy to overcome the obvious limitations of single cell seeding on patterned substrates.

Reviewer #2:

Remarks to the Author:

The authors have extensively revised their manuscript including addition of a number of experiments, such as blocking contraction to evaluate the importance of contractile force on cardiomyocyte maturation. These points largely answer my previous concerns. The revised manuscript documents in detail a new approach to study links between cardiomyocyte function and maturation that could be of broad interest and contribute to enhanced function of stem cell-derived cardiomyocytes.

Reviewer #3:

Remarks to the Author:

The authors have adequately addressed the concerns raised in initial review by adding a considerable amount of new data and clarifying concerns in the text.